# Decentralized multi-agent reinforcement learning via anticipation sharing

## Abstract

Centralized multi-agent reinforcement learning requires global policy access and coordination, often infeasible in decentralized applications. A challenge in decentralized MARL with individual rewards is misaligned local objectives without global coordination. Existing methods that share rewards, values or full policies have high overheads and coupled learning. We introduce a novel decentralized MARL method called Anticipation Sharing that induces coordination by sharing limited policy information. Agents update anticipations of peer action distributions, share these with neighbors, and identify deviations between individual and collective objectives. By exchanging anticipations, agents align behaviors without prohibitive overheads of full policy sharing. Our simulations demonstrate Anticipation Sharing enables decentralized cooperative learning using only local interactions.

## 1 Introduction

Multi-agent reinforcement learning (MARL) enables collaborative decision-making in environments with distributed agents. It has diverse real-world applications including autonomous vehicles, robotics, and communications systems. Centralized MARL requires global information access and a central coordinator, often infeasible in decentralized settings.

Without access to team rewards or objectives, decentralized agents face social dilemmas - prioritizing individual rewards can produce suboptimal collective outcomes. The Prisoner's Dilemma exemplifies this tension. When agents act purely out of self-interest, they achieve lower returns compared to cooperating for the common good (Debreu, 1954). Yet determining optimal collaborative strategies is challenging when only seeing a local viewpoint. Decentralized MARL tackles these cooperation challenges in distributed environments with individual rewards. By developing algorithms that align decentralized policies without global knowledge, agents can learn to optimize collective returns through only local interactions. This addresses real-world coordination problems where central controllers are infeasible.

Several MARL methodologies have recently been proposed to enable decentralized learning, but they attribute a team reward to each agent, which is infeasible when an agent is privy only to its individual reward (Sun et al., 2022; Lauer & Riedmiller, 2000; Boutilier, 1996; Jiang & Lu, 2022). To enhance cooperation among agents while keeping private individual rewards, several methods propose the exchange of information. For instance, some strategies involve sharing rewards to guide agents towards a collective optimum (Chu et al., 2020b; Yi et al., 2022; Chu et al., 2020a). Others suggest sharing value function model parameters or values of the value functions through the aggregation from neighboring agents to achieve similar ends (Zhang et al., 2018a;b; 2020; Suttle et al., 2020; Du et al., 2022). In these approaches, agents calculate a global value based on shared rewards or values, and subsequently, they adjust their policies to maximize this aggregated value. Some studies have explored consensus strategies focusing on policy rather than value (Zhang & Zavlanos, 2019; Stankovic et al., 2022a;b).

In real-world applications, the issue of privacy, particularly concerning rewards and values, becomes a significant hurdle. Agents often prioritize keeping this information confidential, posing a challenge to the practicality of methods that require such sharing. Additionally, sharing model parameters incurs substantial communication overhead and also privacy concerns, which can also result in the transfer of excessive and non-essential information, thereby slowing the learning process.

In this paper, in response to the above challenges, we introduce a novel approach for decentralized cooperative policy learning when agents have individual rewards and no global perspectives. A key advantage of our method is achieving emergent collaboration without sharing sensitive information like actual rewards or model parameters between agents. The core concept we leverage is *anticipation sharing* (AS). Agents share anticipated action distributions, reflecting their preferences. The anticipations to other agents are solved by each agents to maximize its own return and then sent to corresponding agents for them to include as constrains when maximizing their returns. Such anticipations carry the information of individual returns implicitly. By exchanging these peer anticipations iteratively, agents can estimate their impacts on collective preferences while preserving individual privacy.

We establish a theoretical lower bound that quantifies the discrepancy between an agent's individual returns and the global collective returns. This enables formulating a surrogate objective for each agent aligned with the global goal while dependent only on local information. Our proposed decentralized MARL algorithm has agents optimize this surrogate through a dual-clipped policy update approach. It imposes constraints that penalize deviations between an agent's policy and peer anticipated policies. This drives agents to converge not just to optimal local policies, but policies contributing to coordination. The iterative anticipation sharing process is central to enabling this decentralized collaborative learning.

In essence, our method induces emergent cooperative behaviors without exposing sensitive individual rewards or models through a decentralized learning framework. Our empirical investigations reinforce the validity of the AS framework, demonstrating its compatitive performance in specific tasks compared to traditional methods. This establishes AS not only as a theoretically sound but also practically effective avenue for harmonizing individual and collective objectives in decentralized cooperation.

## 2 RELATED WORK

**Centralised learning.** Centralized learning in MARL typically involves a central unit that processes and coordinates actions across all agents. This approach, facilitating a comprehensive view of the environment, enables agents to optimize policies based on collective goals and shared information. Numerous contemporary MARL studies focus on optimizing multi-agent policies under the assumption of an evenly split shared team reward (Kuba et al., 2022; Su & Lu, 2022; Wu et al., 2021). These studies often employ a blend of centralized learning and decentralized execution. For instance, some utilize centralized learning during policy development for optimal coordination, followed by decentralized execution allowing agents to act independently Kuba et al. (2022); Wu et al. (2021). Conversely, others adopt a decentralized learning approach while maintaining shared parameters across networks, a method that navigates between full centralization and independent agent operation (Sun et al., 2022)]. In contrast to these methodologies, our research takes a distinct path by exploring decentralized MARL in environments where each agent operates based on individual rewards, without the reliance on a common team reward. This approach reflects a more realistic scenario in many real-world applications, where agents need to make autonomous decisions based on limited, individual information, and where centralized coordination is either impractical or undesirable due to privacy or scalability concerns.

**Value sharing**. Value sharing methods use shared Q-values or state-values among agents to better align individual and collective goals. Many of these methods utilize consensus techniques to estimate the value of a joint policy and guide individual policy updates accordingly. For instance, a number of networked actor-critic algorithms exist based on value function consensus, wherein agents merge individual value functions towards a global consensus by sharing parameters (Zhang et al., 2018a;b; 2020; Suttle et al., 2020). For communication efficiency, some algorithms reduce the parameters shared (Lin et al., 2019) while others emphasize sharing function values for global value estimation (Du et al., 2022). However, these methods have an inherent limitation: agents modify policies individually, using fixed Q-values or state-values, making them less adaptive to immediate policy shifts from peers, which may introduce policy discoordination. In contrast, our approach enables more adaptive decentralized coordination by having agents directly share and respond to peer policy anticipations.

**Reward sharing**. Reward sharing is about receiving feedback from a broader system-wise outcome perspective, ensuring that agents act in the group's collective best interest. Some works have introduced a spatially discounted reward function (Chu et al., 2020b;a). In these approaches, each agent collaboratively shares rewards within its vicinity. Subsequently, an adjusted reward is derived by amalgamating the rewards of proximate agents, with distance-based discounted weights. Other methods advocate for the dynamic learning of weights integral to reward sharing, which concurrently evolve as agents refine their policies (Yi et al., 2022). In our research, we focus on scenarios where agents know only their individual rewards and are unaware of their peers' rewards. This mirrors real-world situations where rewards are kept confidential or sharing rewards suffers challenges such as communication delays and errors. Consequently, traditional value or reward sharing methods fall short in these contexts. In contrast, our method induces coordination without requiring reward sharing.

**Policy sharing**. Policy sharing strives to unify agents' behaviors through an approximate joint policy. However, crafting a global policy for each agent based on its individual reward can lead to suboptimal outcomes. Consensus update methods offer a solution by merging individually learned policies towards an optimal policy. Several studies have employed such a strategy, focusing on a weighted sum of neighboring agents' policy model parameters (Zhang & Zavlanos, 2019; Stankovic et al., 2022a;b). These methods are particularly useful when sharing individual rewards or value estimates is impractical. Yet, sharing policy model parameters risks added communication overheads and data privacy breaches. Whereas these methods share model parameters directly for policy consensus, we have agents share anticipations of policy outputs, avoiding parameter sharing.

**Social dilemmas.** Social dilemmas highlight the tension between individual pursuits and collective outcomes. In these scenarios, agents aiming for personal gains can lead to compromised group results. For instance, one study has explored self-driven learners in sequential social dilemmas using independent deep Q-learning (Leibo et al., 2017). A prevalent research direction introduces intrinsic rewards to encourage collective-focused policies. For example, *moral learners* have been introduced with varying intrinsic rewards (Tennant et al., 2023) whilst other approaches have adopted an inequity aversion-based intrinsic reward (Hughes et al., 2018) or rewards accounting for social influences and predicting other agents' actions (Jaques et al., 2019). Borrowing from economics, certain methods have integrated formal contracting to motivate global collaboration (Christoffersen et al., 2023). While these methods modify foundational rewards, we maintain the original objectives, emphasizing a collaborative, information-sharing strategy to nurture cooperative agents.

**Teammate modelling** Teammate/opponent modeling in MARL often relies on agents having access to, or inferring, information about teammates' goals, actions, or rewards. This information is then used to improve collective outcomes (Albrecht & Stone, 2018; He et al., 2016). Our approach differs from traditional team modeling. Rather than focusing on predicting teammates' exact actions or strategies, our method involves each agent calculating and sharing anticipated action distributions that would benefit its own strategy. These anticipations are used by other agents (not the agent itself) to balance their own returns with the return of the agent sending the anticipation. This approach emphasizes anticipations that serve the agent's own return optimization. Coordination occurs through strategic adaptation based on *others' anticipations* that implicitly include information about their returns, rather than accurately modeling their behaviors. This key difference highlights our decentralized decision-making and coordination approach. It contrasts with conventional team modeling in MARL that focuses on modeling teammates' behaviors directly.

## 3 BACKGROUND AND PROBLEM STATEMENT

In this work, we approach the collaborative, decentralized multi-agent reinforcement learning problem with individual rewards using Networked Multi-agent Markov Decision Processes (Networked MMDPs). Specifically, we consider a Networked MMDP with $N$ agents, which can be represented as a tuple $< \mathcal{G}, \mathcal{S}, \{\mathcal{A}^i\}_{i=1}^N, \mathcal{P}, \{\mathcal{R}^i\}_{i=1}^N, \gamma >$, where $\mathcal{G} = (\mathcal{V}, \mathcal{E})$ denotes a communication graph, $\mathcal{S}$ denotes a global state space, $\mathcal{A}^i$ is the individual action space, $\mathcal{A} = \Pi_{i=1}^N \mathcal{A}^i$ is the joint action space, $\mathcal{P} : \mathcal{S} \times \mathcal{A} \times \mathcal{S} \to [0, 1]$ is the state transition function, $\mathcal{R}^i : \mathcal{S} \times \mathcal{A} \to \mathbb{R}$ is the individual reward function, and $\gamma$ is a discount factor. Each agent $i$ selects action $a^i \in \mathcal{A}^i$ based on its individual policy $\pi^i : \mathcal{S} \times \mathcal{A}^i \to [0, 1]$. The joint action of all agents is represented by $\boldsymbol{a} \in \mathcal{A}$, and the joint policy across these agents, conditioned on state $s \in \mathcal{S}$, is denoted as $\boldsymbol{\pi}(\cdot|s) = \prod_{i=1}^N \pi^i(\cdot|s)$.

The primary objective in this setting is to maximize the cumulative discounted return for all agents,

$$\eta(\boldsymbol{\pi}) = \sum_{i=1}^{N} \mathbb{E}_{\tau \sim \boldsymbol{\pi}} \left[ \sum_{t=0}^{\infty} \gamma^t r_t^i \right], \tag{1}$$

where the expectation, $\mathbb{E}_{\tau \sim \boldsymbol{\pi}}[\cdot]$, is computed over trajectories with an initial state distribution $s_0 \sim d^{\boldsymbol{\pi}}(s)$, action selection $\boldsymbol{a}_t \sim \boldsymbol{\pi}(\cdot|s_t)$, and state transitions $s_{t+1} \sim \mathcal{P}(\cdot|s_t, \boldsymbol{a}_t)$. The reward for an agent $i$ is $r_t^i = \mathcal{R}^i(s, \boldsymbol{a})$. In our setup, agents must adjust their strategies in situations where rewards might conflict and without access to shared reward information.

An individual advantage function is also introduced,

$$A_i^{\boldsymbol{\pi}}(s, \boldsymbol{a}) = Q_i^{\boldsymbol{\pi}}(s, \boldsymbol{a}) - V_i^{\boldsymbol{\pi}}(s) \tag{2}$$

which depends on the individual state-value and action-value functions, respectively

$$V_i^{\boldsymbol{\pi}}(s) = \mathbb{E}_{\tau \sim \boldsymbol{\pi}} \left[ \sum_{t=0}^{\infty} \gamma^t r_t^i | s_0 = s \right], \quad Q_i^{\boldsymbol{\pi}}(s, \boldsymbol{a}) = \mathbb{E}_{\tau \sim \boldsymbol{\pi}} \left[ \sum_{t=0}^{\infty} \gamma^t r_t^i | s_0 = s, \boldsymbol{a}_0 = \boldsymbol{a} \right]. \tag{3}$$

## 4 METHODOLOGY

In decentralized settings with individual rewards, agents must balance personal objectives with collective goals, despite lacking global perspectives. Our approach, *anticipation sharing* (AS), facilitates this dual awareness without direct reward or objective sharing. Agents exchange anticipations about peer actions solved by maximizing their own return and take the anticipations from others into account when solving policies to maximize individual return, enabling each agent to infer collective objectives. This allows understanding broader impacts of actions through localized interactions.

Unlike traditional methods that share explicit rewards or objectives, AS involves agents exchanging anticipations that implicitly include the information of others' objectives. By observing how its actions align with aggregated anticipations, each agent can perceive the divergence between its individual interests and the inferred collective goals. This drives policy updates to reduce the identified discrepancy, bringing local and global objectives into closer alignment.

Our constrained optimization approach leverages the identified divergences between individual and collective objectives to align decentralized policies. Agents iteratively share anticipated actions and adapt policies accounting for peer anticipations. This fosters continuous, adaptive refinement of strategies balancing both individual returns and collective dynamics inferred from shared anticipations. Our algorithm harnesses this divergence identification, ensuring decision-making integrates individual rewards and collective objectives surmised from interactions.

### 4.1 THEORETICAL DEVELOPMENTS

We commence our technical developments by analyzing joint policy shifts in a centralized setting. This parallels foundational trust region policy optimization work Schulman et al. (2015). We prove the following bound on the expected return difference between new and old joint policies:

**Theorem 1** *We establish a bound for the difference in expected returns between an old joint policy $\boldsymbol{\pi}_{old}$ and a newer policy $\boldsymbol{\pi}_{new}$:*

$$\eta(\boldsymbol{\pi}_{new}) \geq \eta(\boldsymbol{\pi}_{old}) + \zeta_{\boldsymbol{\pi}_{old}}(\boldsymbol{\pi}_{new}) - C \cdot D_{KL}^{max}(\boldsymbol{\pi}_{old}||\boldsymbol{\pi}_{new}), \tag{4}$$

*where*

$$\zeta_{\boldsymbol{\pi}_{old}}(\boldsymbol{\pi}_{new}) = \mathbb{E}_{s \sim d^{\boldsymbol{\pi}_{old}}(s), \boldsymbol{a} \sim \boldsymbol{\pi}_{new}(|s)} \left[ \sum_i A_i^{\boldsymbol{\pi}_{old}}(s, \boldsymbol{a}) \right],$$

$$C = \frac{4 \max_{s, \boldsymbol{a}} |\sum_i A_i^{\boldsymbol{\pi}_{old}}(s, \boldsymbol{a})| \gamma}{(1 - \gamma)^2} \tag{5}$$

$$D_{KL}^{max}(\boldsymbol{\pi}_{old}||\boldsymbol{\pi}_{new}) = \max_s D_{KL}(\boldsymbol{\pi}_{old}(\cdot|s)||\boldsymbol{\pi}_{new}(\cdot|s)).$$

The proof is given in Appendix A.1.

The key insight is that the expected improvement in returns under the new policy depends on both the expected advantages it provides over the old policy, as well as the divergence between policy distributions. This quantifies the impact of joint policy changes on overall system performance given global knowledge, extending trust region concepts to multi-agent domains.

However, this result relies on the strong assumption of centralized execution with full observability of joint policies. To address this limitation, we introduce the concept of an *anticipated joint policy* from each agent's local perspective. As we will show, the anticipated joint policy is solved by optimizing individual objectives. Analyzing anticipated policies is crucial for assessing the discrepancy between individual objectives and the original collective one in decentralized learning.

**Definition 1** *For each agent in a multi-agent system, we define the **anticipated joint policy**, denoted as $\tilde{\boldsymbol{\pi}}^i$, formulated as $\tilde{\boldsymbol{\pi}}^i(\boldsymbol{a}|s) = \prod_{j=1}^{N} \pi^{ij}(a^j|s)$. Here, for each agent $i$, $\pi^{ij}$ represents the anticipation from agent $i$ to agent $j$'s policy when $j \neq i$. When $j = i$, we use $\pi^{ii} = \pi^i$ to indicate agent $i$'s own policy. To represent the collection of all such anticipated joint policies across agents, we use the notation $\tilde{\boldsymbol{\Pi}} := (\tilde{\boldsymbol{\pi}}^1, \cdots, \tilde{\boldsymbol{\pi}}^i, \cdots, \tilde{\boldsymbol{\pi}}^N)$.*

The anticipated joint policy represents an agent's perspective of the collective strategy constructed from its own policy and anticipations to peers. We will present how to solve such anticipated joint policy in Section 4.2.

**Definition 2** *The **total expectation of individual advantages**, considering the anticipated joint policies and a common state distribution, is defined as follows:*

$$\zeta_{\boldsymbol{\pi}'}(\tilde{\boldsymbol{\Pi}}) = \sum_i \mathbb{E}_{s \sim d^{\boldsymbol{\pi}'}(s), \boldsymbol{a} \sim \tilde{\boldsymbol{\pi}}^i(\boldsymbol{a}|s)} \left[ A_i^{\boldsymbol{\pi}'}(s, \boldsymbol{a}) \right], \tag{6}$$

*where $\zeta_{\boldsymbol{\pi}'}(\tilde{\boldsymbol{\Pi}})$ represents the sum of expected advantages for each agent $i$, calculated over their anticipated joint policy $\tilde{\boldsymbol{\pi}}^i$ and a shared state distribution, $d^{\boldsymbol{\pi}'}(s)$. The advantage $A_i^{\boldsymbol{\pi}'}(s, \boldsymbol{a})$ for each agent is evaluated under a potential joint policy $\boldsymbol{\pi}'$, which may differ from the actual joint policy $\boldsymbol{\pi}$ in play. This definition captures the expected benefit each agent anticipates based on the anticipated joint actions, relative to the potential joint policy $\boldsymbol{\pi}'$.*

This concept quantifies the expected cumulative advantage an agent could hypothetically gain by switching from some reference joint policy to the anticipated joint policies of all agents. It encapsulates the perceived benefit of the anticipated decentralized policies versus a centralized benchmark. Intuitively, if an agent's anticipations are close to the actual policies of other agents, this expected advantage will closely match the actual gains. However, discrepancies in anticipations will lead to divergences, providing insights into the impacts of imperfect decentralized knowledge.

Equipped with these notions of anticipated joint policies and total advantage expectations, we can analyze the discrepancy of the expectation of the total advantage caused by policy shift from the actual joint policy to the individually anticipated ones. Specifically, we prove the following bound relating this discrepancy:

**Theorem 2** *The discrepancy between $\zeta_{\boldsymbol{\pi}'}(\tilde{\boldsymbol{\Pi}})$ and $\zeta_{\boldsymbol{\pi}'}(\boldsymbol{\pi})$ is upper bounded as follows:*

$$\zeta_{\boldsymbol{\pi}'}(\tilde{\boldsymbol{\Pi}}) - \zeta_{\boldsymbol{\pi}'}(\boldsymbol{\pi}) \leq f^{\boldsymbol{\pi}'} + \sum_i \frac{1}{2} \max_{s, \boldsymbol{a}} \left| A_i^{\boldsymbol{\pi}'}(s, \boldsymbol{a}) \right| \cdot \sum_{s, \boldsymbol{a}} \left( \tilde{\boldsymbol{\pi}}^i(\boldsymbol{a}|s) - \boldsymbol{\pi}(\boldsymbol{a}|s) \right)^2, \tag{7}$$

*where*

$$f^{\boldsymbol{\pi}'} = \sum_i \frac{1}{2} \max_{s, \boldsymbol{a}} \left| A_i^{\boldsymbol{\pi}'}(s, \boldsymbol{a}) \right| \cdot |\mathcal{A}| \cdot \|d^{\boldsymbol{\pi}'}\|_2^2 \tag{8}$$

The proof is given in Appendix A.2.

This result quantifies the potential drawbacks of relying on imperfect knowledge in decentralized settings, where agents' anticipations may diverge from actual peer policies. It motivates reducing the difference between anticipated and actual policies.

Previous results bounded the deviation between total advantage expectations under the actual joint policy versus under anticipated joint policies. We now build on this to examine how relying too much on past policies can lead to misjudging the impact of new joint policy shifts over time. Specifically, we consider the relationship between $\zeta_{\pi_{\text{old}}}(\tilde{\mathbf{\Pi}}_{\text{new}})$, the perceived benefit of the new anticipated joint policies $\tilde{\mathbf{\Pi}}_{\text{new}}$, assessed from the perspective of the previous joint policy $\pi_{\text{old}}$, and $\eta(\pi_{new})$, which measures the performance of the new joint policy. The former represents a potentially myopic perspective informed heavily by the past policy and, as such, it may inaccurately judge the actual impact of switching to $\pi_{\text{new}}$ as quantified by $\eta(\pi_{\text{new}})$. The following result provides a lower bound of the expected return, $\eta(\pi_{new})$, of the newer joint policy.

**Theorem 3** *The expected return of the newer joint policy is lower bounded as follows:*

$$\eta(\pi_{new}) \geq \zeta_{\pi_{old}}(\tilde{\mathbf{\Pi}}_{new}) + \eta(\pi_{old}) - C \cdot \sum_i D_{KL}^{max}(\pi_{old}^{ii} || \pi_{new}^{ii})$$
$$- f^{\pi_{old}} - \sum_i \frac{1}{2} \max_{s,\boldsymbol{a}} |A_i^{\pi_{old}}(s,\boldsymbol{a})| \cdot \sum_{s,\boldsymbol{a}} \left( \tilde{\pi}_{new}^i(\boldsymbol{a}|s) - \pi_{new}(\boldsymbol{a}|s) \right)^2 . \tag{9}$$

The full proof is given in Appendix A.3.

This theorem explains the nuanced dynamics of policy changes in decentralized multi-agent reinforcement learning, where agents learn separately. It sheds light on how uncoordinated local updates between individual agents affect the collective performance. At the same time, this result suggests a potential way to improve overall performance by leveraging the anticipated joint policies held by each agent.

## 4.2 A SURROGATE OPTIMIZATION OBJECTIVE

Our preceding results established analytical foundations for assessing joint policy improvement in such settings. We now build upon these results to address the practical challenge of how agents can effectively optimize system-wide returns in a decentralized fashion.

Directly maximizing the expected collective returns, $\eta(\pi)$ is intractable without a global view. However, Theorem 3 provides the insight that agents can optimize a more tractable localized surrogate objective, $\zeta_{\pi_{\text{old}}}(\tilde{\mathbf{\Pi}})$. This simplifies the global objective into a decentralized form dependent only on an agent's individual policy, denoted as $\pi^{ii}$, and its anticipations to others, $\pi^{ij}$, retaining the relevant complexities in a decentralized form. To this end, instead of using the original global objective, we leverage the lower bound given by Theorem 3: by maximizing the lower bound, the collective return can be maximized. Since the terms $\eta(\pi_{old})$ and $f^{\pi_{\text{old}}}$ featuring in Theorem 3 are not relevant to optimizing $\tilde{\mathbf{\Pi}}$, they can be been omitted. Thus, we propose the following global constrained optimization problem as a surrogate objective of the original collective one:

$$\max_{\tilde{\mathbf{\Pi}}} \zeta_{\pi_{old}}(\tilde{\mathbf{\Pi}})$$
$$\text{s.t.} \quad \sum_i D_{KL}^{max}(\pi_{old}^{ii} || \pi^{ii}) \leq \delta, \quad \sum_i \max_{s,\boldsymbol{a}} |A_i^{\pi_{old}}(s,\boldsymbol{a})| \cdot \sum_{s,\boldsymbol{a}} \left( \tilde{\pi}^i(\boldsymbol{a}|s) - \pi(\boldsymbol{a}|s) \right)^2 \leq \delta'. \tag{10}$$

This global optimization objective captures the essence of coordinating joint policies to maximize localized advantages. However, it still assumes a centralized executor with full knowledge of $\tilde{\mathbf{\Pi}}$.

To make this feasible in decentralized MARL, we reformulate it from each agent's limited perspective. Remarkably, we can distill the relevant components into a local objective and constraints for each individual agent, as follows:

$$\max_{\tilde{\pi}^i} \mathbb{E}_{s \sim d_{old}^{\pi}(s), \boldsymbol{a} \sim \tilde{\pi}^i(\boldsymbol{a}|s)} \left[ A_i^{\pi_{old}}(s,\boldsymbol{a}) \right]$$
$$\text{s.t.} : \quad \text{(a)} \quad D_{KL}^{max}(\pi_{old}^{ii} || \pi^{ii}) \leq \delta_1, \quad \text{(b)} \quad \kappa_i \cdot \sum_{s,a_j} (\pi^{ij}(a_j|s) - \pi^{jj}(a_j|s))^2 \leq \delta_2, \ \forall j \neq i, \tag{11}$$
$$\text{(c)} \quad \kappa_i \cdot \sum_{s,a_i} (\pi^{ii}(a_i|s) - \pi^{ji}(a_i|s))^2 \leq \delta_2, \ \forall j \neq i,$$

where $\kappa_i = \max_{s,\boldsymbol{a}} |A_i^{\boldsymbol{\pi}_{old}}(s,\boldsymbol{a})|$.

Note that the constraints in Eq. 11 depend on other agents' policies $\pi^{jj}$ as well as their anticipations of agent $i$'s policy, $\pi^{ji}$. To evaluate these terms, each agent $j$ needs to share its action distribution $\pi^{jj}(\cdot|s)$ and the anticipated action distribution $\pi^{ji}(\cdot|s)$ to agent $i$. This sharing allows each agent $i$ to assess the constraint terms, which couple the individual advantage optimizations under local constraints. Such constraints reflect not only the differences between the true policy of others and the anticipations to them from an agent, but also the discrepancy between the agent's own true policy and the anticipations from others. Distributing the optimization while exchanging critical policy information in this way balances autonomy for decentralized execution with maintaining global coordination between agents.

This setup differs from teammate modeling where agent $i$ tries to approximate peer policies $\hat{\pi}^{ij}$ and use them when solving $\pi^{ii}$, whereas Eq. 11 aims to optimize the anticipations $\pi^{ij}$ together with $\pi^{ii}$ and then $\pi^{ij}$ is used by agent $j$ to solve $\pi^{jj}$. Therefore, the anticipations include the information about individual objectives implicitly. By exchanging the anticipations, individual agents can balance others' objectives and thus the collective performance when optimizing its own objective. This setup also significantly differs from fully centralized learning where a coordinator has access to all policies. Here agents only share action distributions to evaluate coupling constraints, retaining decentralized computation.

## 4.3 A PRACTICAL ALGORITHM FOR LEARNING WITH AS

We propose a structured approach to optimize the objective in Eq. 11. The derivation of the algorithm involves specific steps, each targeting different aspects of the optimization challenge. Note that in this practical algorithm, we present a general setup where the network topology of the system does not need to be fully-connected. Each agent only exchanges information with neighbours $\{j | j \in \mathcal{N}_i\}$. This provides an approximation of the theoretical results.

**Step 1: Clipping Policy Ratio for KL Constraint.**   Addressing the KL divergence constraint (a) in Eq. 11 is crucial in ensuring our decentralized learning process remains effective. This constraint ensures that updates to an agent's individual policy do not deviate excessively from its previous policy. To manage this, we incorporate a clipping mechanism, inspired by PPO-style clipping (Schulman et al., 2017), adapted for decentralized agents.

We start by defining probability ratios for the individual policy and anticipated peer policies:

$$\xi_i = \frac{\pi^{ii}(a_i|s';\theta^{ii})}{\pi_{old}^{ii}(a_i|s';\theta_{old}^{ii})}, \quad \xi_{\mathcal{N}_i} = \prod_{j \in \mathcal{N}_i} \frac{\pi^{ij}(a_j|s;\theta^{ij})}{\pi_{old}^{jj}(a_j|s;\theta_{old}^{jj})}. \tag{12}$$

These ratios measure the extent of change in an agent's policy relative to its previous one and its anticipations to others. We then apply a clipping operation to $\xi_i$, the individual policy ratio:

$$\mathbb{E}_{s \sim d^{\boldsymbol{\pi}_{old}}(s), \boldsymbol{a} \sim \boldsymbol{\pi}_{old}(\boldsymbol{a}|s)} \left[ \min \left( \xi_i \xi_{\mathcal{N}_i} \hat{A}_i, \text{clip}(\xi_i, 1-\epsilon, 1+\epsilon) \xi_{\mathcal{N}_i} \hat{A}_i \right) \right].$$

This method selectively restricts major changes to the individual policy $\pi^{ii}$, while allowing more flexibility in updating anticipations of peer policies. It balances the adherence to the KL constraint with the flexibility needed for effective learning and adaptation in a decentralized environment.

**Step 2: Penalizing Anticipation Discrepancies.**   The objective of this step is to enforce constraints (b) and (c) in Eq. 11, which aim to penalize discrepancies between the anticipated and actual policies. Simply optimizing the advantage function may not sufficiently increase these discrepancies. Therefore, we introduce penalty terms that are activated when policy updates inadvertently increase these discrepancies. Specifically, we define states $X^{ij}$ to identify when the policy update driven by the advantage exacerbates the discrepancies between the resulting anticipated policies and other agents' current policies, and $X^{ii}$ to identify the discrepancies between the resulting agent's own policy and the ones anticipated from other agents. These are defined as

$$X^{ij} = \left\{ (s,\boldsymbol{a}) \mid \frac{\pi^{ij}(a_j|s;\theta^{ij})}{\pi^{jj}(a_j|s)} \hat{A}_i > \hat{A}_i \right\}, \qquad X^{ii} = \left\{ (s,\boldsymbol{a}) \mid \frac{\pi^{ii}(a_i|s;\theta^{ii})}{\pi^{ji}(a_i|s)} \hat{A}_i > \hat{A}_i \right\}, \tag{13}$$

where the pairs $(s, \boldsymbol{a})$ represent scenarios in which the gradient influenced by $\hat{A}_i$ increases the divergence between the two policies. The following indicator function captures this effect:

$$\mathbb{I}_X(s, \boldsymbol{a}) = \begin{cases} 1 & \text{if } (s, \boldsymbol{a}) \in X, \\ 0 & \text{otherwise.} \end{cases} \tag{14}$$

**Step 3: Dual Clipped Objective.** In the final step, we combine the clipped surrogate objective with coordination penalties to form our dual clipped objective:

$$\max_{\theta^{ii}, \boldsymbol{\theta}_{-ii}} \mathbb{E}_{s \sim d^{\boldsymbol{\pi}_{old}(s)}, \boldsymbol{a} \sim \boldsymbol{\pi}_{old}(\boldsymbol{a}|s)} [\min \left( \xi_i \xi_{\mathcal{N}_i} \hat{A}_i, \text{clip}(\xi_i, 1 - \epsilon, 1 + \epsilon) \xi_{\mathcal{N}_i} \hat{A}_i \right)$$
$$- \kappa_i \cdot \sum_{j \in \mathcal{N}_i} \rho_j \mathbb{I}_{X^{ij}}(s, \boldsymbol{a}) \| \pi^{ij}(\cdot|s; \theta^{ij}) - \pi^{jj}(\cdot|s) \|_2^2 + \rho'_j \mathbb{I}_{X^{ii}}(s, \boldsymbol{a}) \| \pi^{ii}(\cdot|s; \theta^{ii}) - \pi^{ji}(\cdot|s) \|_2^2]. \tag{15}$$

This step balances individual policy updates with the need for coordination among agents, thereby aligning individual objectives with collective goals.

**Implementation details.** In our implementation, we use $\hat{\kappa}_i = \text{mean}_{s,\boldsymbol{a}} |\hat{A}_i^{\boldsymbol{\pi}}|$ to approximate $\kappa_i$ in order to mitigate the impact of value overestimation. Additionally, we adopt the same value for the coefficients $\rho_j$ and $\rho'_j$ across different $j$, and denote it as $\rho$. We also utilize the generalized advantage estimator (GAE) Schulman et al. (2016) due to its well-known properties to obtain estimates

$$\hat{A}_i^t = \sum_{l=0}^{\infty} (\gamma \lambda)^l \delta_{t+l}^{V_i}, \qquad \delta_{t+l}^{V_i} = r_i^{t+l} + \gamma V_i(s_{t+l+1}) - V_i(s_{t+l}), \tag{16}$$

where $V_i$ is approximated by minimizing the following loss function,

$$\mathcal{L}_{V_i} = \mathbb{E}[(V_i(s_t) - \sum_{l=0}^{\infty} \gamma^l r_i^{t+l})^2]. \tag{17}$$

Algorithm 1 in Appendix. F presents the detailed procedure used in our experimental section. Appendix. E shows an illustration of our method.

## 5 EXPERIMENTS

### 5.1 TASKS AND BASELINES

We evaluate the performance of our AS algorithm across a spectrum of tasks, spanning both discrete (Exchange and Cooperative Navigation) and continuous (Cooperative Predation) spaces and featuring diverse agent counts (from 3 to 20 agents). For a comprehensive assessment, we benchmark AS against three prominent baselines: Value Sharing (VS) Du et al. (2022), Value Function Parameter Sharing (VPS) Zhang et al. (2018b), and Policy Model Sharing (PS) Zhang & Zavlanos (2019). A detailed description of the environments and baselines can be found in the Appendix.

### 5.2 RESULTS

The training curves and final total returns of the different algorithms are shown in Figure 1. For the two discrete environments, "Exc." and "Navi.", there are 3 agents. The neighboring agents of each agent are enclosed within the dashed outline rectangles, as depicted in Figures 2(a) and (b) in Appendix B. In the continuous domain, we assess the algorithms using 6, 8, and 12 agents. Neighboring agents are defined as those within a normalized distance of 0.1. For each algorithm and task, we conduct 5 runs with different seeds. As seen in Figure 1, our AS algorithm performs the best consistently across all tasks, attaining policies that gain more total return than the baselines. This demonstrates the effectiveness and superiority of AS. It is important to note that the aim of our study is not to outperform the baseline algorithms but to provide a viable alternative in settings where agents cannot exchange values or rewards due to privacy constraints.

For the baseline algorithms, VS and VPS exhibit unstable performance across tasks. This implies merely sharing values or value functions and achieving value consensus may be insufficient for cooperative policies. A hypothesis for the performance disparities is that despite approximating

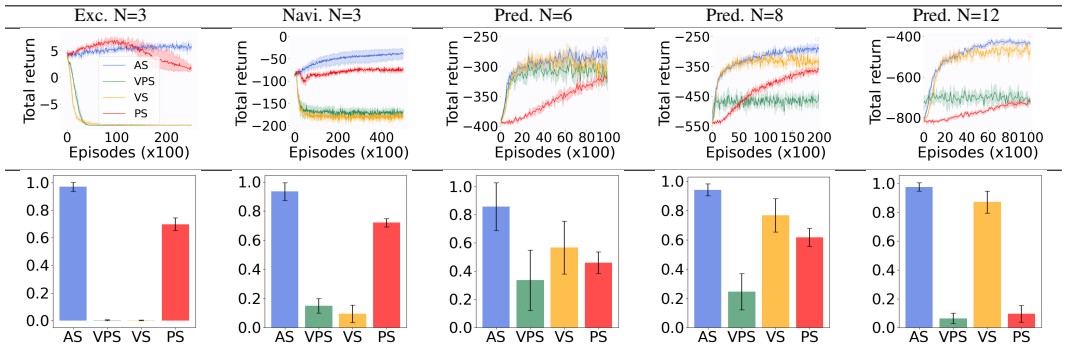

Figure 1: Training curves (top row) and normalized final total return (bottom row).

system-wide values, policy updates in these methods lack coordination, leading to inferior cooperation. Particularly in the Pred. task, VS and VPS exhibit better performance in some scenarios compared to Exc. and Navi. tasks. This difference can be explained by the nature of the tasks themselves. Exc. and Navi. demand a higher level of coordination, especially because agents are heterogeneous with unique individual objectives. Such environments intensify the need for precise and synchronized policy updates, making the coordination challenge more pronounced. In contrast, our method aims to address this discoordination by enabling more harmonized policy updates among agents, taking into consideration the anticipations of other agents' policies, which leads to a more cohesive policy development process. PS also focuses on direct policy coordination rather than value consensus. However, results show PS has slow convergence on some tasks. Sharing policy parameters may entail redundant information unnecessary for effective coordination. In contrast, AS avoids sharing policy parameters, instead exchanging action distributions from policies. Furthermore, in AS each agent selectively shares anticipations only with corresponding agents, not indiscriminately with all neighbors. Our superior training efficiency and performance compared to PS showcases this benefit. As agent populations increase, PS convergence slows, while AS remains robust.

We also conducted further studies regarding the scalability, impact of neighbourhood range, sensitivity to the penalty weight. Experimental results indicate AS's robust performance with sparse network topology, different neighbour counts, and varying penalty weigh. Details are given in Appendix.

## 6    CONCLUSIONS AND FUTURE WORK

In this work, we tackled the challenge of decentralized multi-agent policy optimization under individual reward conditions, where individual interests can conflict with collective objectives. We introduced Anticipation Sharing (AS) as an alternative to traditional methods like intrinsic rewards, value sharing, and policy model sharing. AS enables agents to incorporate their individual interests into anticipations regarding the action distributions of other agents. In the process of exchanging their anticipations with each other, agents become aware of the collective interest implicitly, despite the fact that rewards, values, and policies are private to each agent.

Theoretically, we established that the difference between agents' actual action distributions and the anticipations from others bounds the difference between individual and collective objectives. We used this insight to create a novel individual objective that serves as a lower bound for the original collective objective, driving agents toward cooperative behaviors. Our decentralized MARL algorithm based on AS demonstrated the capability to produce pro-social agents in empirical experiments.

In the future, several opportunities exist to enhance our understanding and application of the AS framework. We can refine individual objectives by investigating tighter bounds for measuring discrepancies between individual and collective interests, and delve deeper into alternative optimization strategies based on AS framework. Another prospective avenue involves exploring the integration of additional communication mechanisms into AS. It would be especially insightful to study these mechanisms within the context of dynamic topology structures that dictate cooperative information flows. Additionally, a thorough analysis of our algorithm's convergence properties would be insightful. Lastly, applying our methodology to more complex tasks remains a promising direction.

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

# A PROOFS

## A.1 PROOF OF THEOREM 1

**Theorem 1** The following bound holds for the difference between the expected returns of the current policy $\boldsymbol{\pi}_{old}$ and another policy $\boldsymbol{\pi}_{new}$

$$\eta(\boldsymbol{\pi}_{new}) \geq \eta(\boldsymbol{\pi}_{old}) + \zeta_{\boldsymbol{\pi}_{old}}(\boldsymbol{\pi}_{new}) - C \cdot D_{KL}^{max}(\boldsymbol{\pi}_{old}||\boldsymbol{\pi}_{new}), \tag{18}$$

where

$$\zeta_{\boldsymbol{\pi}_{old}}(\boldsymbol{\pi}_{new}) = \mathbb{E}_{s \sim d^{\boldsymbol{\pi}_{old}}(s), \boldsymbol{a} \sim \boldsymbol{\pi}_{new}(\cdot|s)} \left[ \sum_i A_i^{\boldsymbol{\pi}_{old}}(s, \boldsymbol{a}) \right],$$

$$C = \frac{4 \max_{s,\boldsymbol{a}} |\sum_i A_i^{\boldsymbol{\pi}_{old}}(s, \boldsymbol{a})| \gamma}{(1 - \gamma)^2} \tag{19}$$

$$D_{KL}^{max}(\boldsymbol{\pi}_{old}||\boldsymbol{\pi}_{new}) = \max_s D_{KL}(\boldsymbol{\pi}_{old}(\cdot|s)||\boldsymbol{\pi}_{new}(\cdot|s)).$$

**Lemma 1** *Given two joint policies $\boldsymbol{\pi}_{old}$ and $\boldsymbol{\pi}_{new}$,*

$$\eta(\boldsymbol{\pi}_{new}) = \eta(\boldsymbol{\pi}_{old}) + \mathbb{E}_{\tau \sim \boldsymbol{\pi}_{new}} \left[ \sum_{i=1}^N \sum_{t=0}^\infty \gamma^t A_i^{\boldsymbol{\pi}_{old}}(s_t, \boldsymbol{a}_t) \right], \tag{20}$$

where $\mathbb{E}_{\tau \sim \boldsymbol{\pi}_{new}}[\cdot]$ means the expectation is computed over trajectories where the initial state distribution $s_0 \sim d^{\boldsymbol{\pi}_{new}}(s)$, action selection $\boldsymbol{a}_t \sim \boldsymbol{\pi}_{new}(\cdot|s_t)$, and state transitions $s_{t+1} \sim \mathcal{P}(\cdot|s_t, \boldsymbol{a}_t)$.

*Proof:* The expected discounted reward of the joint policy, i.e., Eq. 1, can be expressed as

$$\eta(\boldsymbol{\pi}) = \sum_{i=1}^N \mathbb{E}_{s \sim d^{\boldsymbol{\pi}}(s)} \left[ V_i^{\boldsymbol{\pi}}(s) \right]. \tag{21}$$

Using $A_i^{\boldsymbol{\pi}_{old}}(s_t, \boldsymbol{a}_t) = \mathbb{E}_{s'}[r_t^i + \gamma V_i^{\boldsymbol{\pi}_{old}}(s') - V_i^{\boldsymbol{\pi}_{old}}(s)]$, we have

$$
\begin{aligned}
& \mathbb{E}_{\tau \sim \boldsymbol{\pi}_{new}} \left[ \sum_{i=1}^N \sum_{t=0}^\infty \gamma^t A_i^{\boldsymbol{\pi}_{old}}(s_t, \boldsymbol{a}_t) \right] \\
& = \mathbb{E}_{\tau \sim \boldsymbol{\pi}_{new}} \left[ \sum_{i=1}^N \sum_{t=0}^\infty \gamma^t (r_t^i + \gamma V_i^{\boldsymbol{\pi}_{old}}(s_{t+1}) - V_i^{\boldsymbol{\pi}_{old}}(s_t)) \right] \\
& = \mathbb{E}_{\tau \sim \boldsymbol{\pi}_{new}} \left[ \sum_{i=1}^N (-V_i^{\boldsymbol{\pi}_{old}}(s_0) + \sum_{t=0}^\infty \gamma^t r_t^i) \right] \\
& = - \sum_{i=1}^N \mathbb{E}_{s_0}[V_i^{\boldsymbol{\pi}_{old}}(s_0)] + \sum_{i=1}^N \mathbb{E}_{\tau \sim \boldsymbol{\pi}_{new}} \left[ \sum_{t=0}^\infty \gamma^t r_t^i \right] \\
& = -\eta(\boldsymbol{\pi}_{old}) + \eta(\boldsymbol{\pi}_{new}).
\end{aligned}
\tag{22}
$$

Thus, we have Eq. 20.

Define an expected joint advantage $\bar{A}_{joint}$ as

$$\bar{A}_{joint}(s) = \mathbb{E}_{\boldsymbol{a} \sim \boldsymbol{\pi}_{new}(\cdot|s)} \left[ \sum_{i=1}^N A_i^{\boldsymbol{\pi}_{old}}(s, \boldsymbol{a}) \right]. \tag{23}$$

Define $L_{\boldsymbol{\pi}_{old}}(\boldsymbol{\pi}_{new})$ as

$$L_{\boldsymbol{\pi}_{old}}(\boldsymbol{\pi}_{new}) = \eta(\boldsymbol{\pi}_{old}) + \sum_s P(s_t = s|\boldsymbol{\pi}_{old}) \sum_{t=0}^\infty \gamma^t \bar{A}_{joint}(s_t). \tag{24}$$

Leveraging the Lemma 2, Lemma 3, and Theorem 1 provided by TRPO Schulman et al. (2015), we have

$$|\eta(\boldsymbol{\pi}_{new}) - L_{\boldsymbol{\pi}_{old}}(\boldsymbol{\pi}_{new})| \le C \cdot (\max_s D_{TV}(\boldsymbol{\pi}_{old}(\cdot|s)||\boldsymbol{\pi}_{new}(\cdot|s)))^2. \tag{25}$$

Based on the relationship: $(D_{TV}(p||q))^2 \le D_{KL}(q||q)$, we have

$$|\eta(\boldsymbol{\pi}_{new}) - L_{\boldsymbol{\pi}_{old}}(\boldsymbol{\pi}_{new})| \le C \cdot D_{KL}^{max}(\boldsymbol{\pi}_{old}||\boldsymbol{\pi}_{new}). \tag{26}$$

For the second term of the RHS of Eq. 24, we have the following equivalent form

$$\begin{aligned}
&\sum_s P(s_t = s|\boldsymbol{\pi}_{old}) \sum_{t=0}^{\infty} \gamma^t \bar{A}_{joint}(s_t) \\
&= \sum_s \sum_{t=0}^{\infty} \gamma^t P(s_t = s|\boldsymbol{\pi}_{old}) \bar{A}_{joint}(s_t) \\
&= \sum_s d^{\boldsymbol{\pi}_{old}}(s) \bar{A}_{joint}(s_t) \\
&= \sum_s d^{\boldsymbol{\pi}_{old}}(s) \mathbb{E}_{\boldsymbol{a} \sim \boldsymbol{\pi}_{new}(\cdot|s)} \left[ \sum_{i=1}^N A_i^{\boldsymbol{\pi}_{old}}(s, \boldsymbol{a}) \right] \\
&= \zeta_{\boldsymbol{\pi}_{old}}(\boldsymbol{\pi}_{new}).
\end{aligned} \tag{27}$$

Thus, we have $L_{\boldsymbol{\pi}_{old}}(\boldsymbol{\pi}_{new}) = \eta(\boldsymbol{\pi}_{old}) + \zeta_{\boldsymbol{\pi}_{old}}(\boldsymbol{\pi}_{new})$. Then, replacing $L_{\boldsymbol{\pi}_{old}}(\boldsymbol{\pi}_{new})$ in Eq. 26, we have

$$|\eta(\boldsymbol{\pi}_{new}) - (\eta(\boldsymbol{\pi}_{old}) + \zeta_{\boldsymbol{\pi}_{old}}(\boldsymbol{\pi}_{new}))| \le C \cdot D_{KL}^{max}(\boldsymbol{\pi}_{old}||\boldsymbol{\pi}_{new}), \tag{28}$$

and thus Theorem 1 is proved.

## A.2 PROOF OF THEOREM 2

**Theorem 2** *The discrepancy between $\zeta_{\boldsymbol{\pi}'}(\tilde{\boldsymbol{\Pi}})$ and the sum of the expected individual advantages calculated with policy $\boldsymbol{\pi}'$ over the true joint policy $\boldsymbol{\pi}$, i.e., $\zeta_{\boldsymbol{\pi}'}(\boldsymbol{\pi})$, is upper bounded as follows.*

$$\zeta_{\boldsymbol{\pi}'}(\tilde{\boldsymbol{\Pi}}) - \zeta_{\boldsymbol{\pi}'}(\boldsymbol{\pi}) \le f^{\boldsymbol{\pi}'} + \sum_i \frac{1}{2} \max_{s,\boldsymbol{a}} \left| A_i^{\boldsymbol{\pi}'}(s,\boldsymbol{a}) \right| \cdot \sum_{s,\boldsymbol{a}} \left( \tilde{\boldsymbol{\pi}}^i(\boldsymbol{a}|s) - \boldsymbol{\pi}(\boldsymbol{a}|s) \right)^2, \tag{29}$$

*where*

$$f^{\boldsymbol{\pi}'} = \sum_i \frac{1}{2} \max_{s,\boldsymbol{a}} \left| A_i^{\boldsymbol{\pi}'}(s,\boldsymbol{a}) \right| \cdot |\mathcal{A}| \cdot \|d^{\boldsymbol{\pi}'}\|_2^2 \tag{30}$$

*Proof:*

$$\begin{aligned}
\zeta_{\boldsymbol{\pi}'}(\tilde{\boldsymbol{\Pi}}) - \zeta_{\boldsymbol{\pi}'}(\boldsymbol{\pi}) &= \sum_i \mathbb{E}_{s \sim d^{\boldsymbol{\pi}'}(s), \boldsymbol{a} \sim \tilde{\boldsymbol{\pi}}^i(\boldsymbol{a}|s)} \left[ A_i^{\boldsymbol{\pi}'}(s,\boldsymbol{a}) \right] - \mathbb{E}_{s \sim d^{\boldsymbol{\pi}'}(s), \boldsymbol{a} \sim \boldsymbol{\pi}(\boldsymbol{a}|s)} \left[ A_i^{\boldsymbol{\pi}'}(s,\boldsymbol{a}) \right] \\
&= \sum_i \sum_{s,\boldsymbol{a}} d^{\boldsymbol{\pi}'}(s)(\tilde{\boldsymbol{\pi}}^i(\boldsymbol{a}|s) - \boldsymbol{\pi}(\boldsymbol{a}|s)) A_i^{\boldsymbol{\pi}'}(s,\boldsymbol{a}), \\
&\le \sum_i \max_{s,\boldsymbol{a}} \left| A_i^{\boldsymbol{\pi}'}(s,\boldsymbol{a}) \right| \cdot \left| \sum_{s,\boldsymbol{a}} d^{\boldsymbol{\pi}'}(s) \left( \tilde{\boldsymbol{\pi}}^i(\boldsymbol{a}|s) - \boldsymbol{\pi}(\boldsymbol{a}|s) \right) \right| \\
&\le \sum_i \max_{s,\boldsymbol{a}} \left| A_i^{\boldsymbol{\pi}'}(s,\boldsymbol{a}) \right| \cdot \sum_{s,\boldsymbol{a}} \frac{1}{2} \left( d^{\boldsymbol{\pi}'}(s)^2 + \left( \tilde{\boldsymbol{\pi}}^i(\boldsymbol{a}|s) - \boldsymbol{\pi}(\boldsymbol{a}|s) \right)^2 \right) \\
&= \sum_i \frac{1}{2} \max_{s,\boldsymbol{a}} \left| A_i^{\boldsymbol{\pi}'}(s,\boldsymbol{a}) \right| \cdot \sum_{s,\boldsymbol{a}} \left( d^{\boldsymbol{\pi}'}(s)^2 + \left( \tilde{\boldsymbol{\pi}}^i(\boldsymbol{a}|s) - \boldsymbol{\pi}(\boldsymbol{a}|s) \right)^2 \right) \\
&= \sum_i \frac{1}{2} \max_{s,\boldsymbol{a}} \left| A_i^{\boldsymbol{\pi}'}(s,\boldsymbol{a}) \right| \cdot \left( |\mathcal{A}| \cdot \|d^{\boldsymbol{\pi}'}\|_2^2 + \sum_{s,\boldsymbol{a}} \left( \tilde{\boldsymbol{\pi}}^i(\boldsymbol{a}|s) - \boldsymbol{\pi}(\boldsymbol{a}|s) \right)^2 \right) \\
&= f^{\boldsymbol{\pi}'} + \sum_i \frac{1}{2} \max_{s,\boldsymbol{a}} \left| A_i^{\boldsymbol{\pi}'}(s,\boldsymbol{a}) \right| \cdot \sum_{s,\boldsymbol{a}} \left( \tilde{\boldsymbol{\pi}}^i(\boldsymbol{a}|s) - \boldsymbol{\pi}(\boldsymbol{a}|s) \right)^2
\end{aligned} \tag{31}$$

where

$$f^{\boldsymbol{\pi}'} = \sum_i \frac{1}{2} \max_{s,\boldsymbol{a}} \left| A_i^{\boldsymbol{\pi}'}(s,\boldsymbol{a}) \right| \cdot |\mathcal{A}| \cdot \|d^{\boldsymbol{\pi}'}\|_2^2.$$

## A.3   PROOF OF THEOREM 3

**Theorem 3** *The expected return of the newer joint policy is lower bounded as follows:*

$$
\begin{aligned}
\eta(\boldsymbol{\pi}_{new}) \geq & \zeta_{\boldsymbol{\pi}_{old}}(\tilde{\boldsymbol{\Pi}}_{new}) + \eta(\boldsymbol{\pi}_{old}) - C \cdot \sum_i D_{KL}^{max}(\pi_{old}^{ii}||\pi_{new}^{ii}) - \\
& \sum_i \frac{1}{2} \max_{s,\boldsymbol{a}} |A_i^{\boldsymbol{\pi}_{old}}(s,\boldsymbol{a})| \cdot \sum_{s,\boldsymbol{a}} \max_s d^{\boldsymbol{\pi}_{old}}(s)^2 + (\tilde{\boldsymbol{\pi}}_{new}^i(\boldsymbol{a}|s) - \boldsymbol{\pi}_{new}(\boldsymbol{a}|s))^2
\end{aligned}
\tag{32}
$$

*Proof* According to Theorem 1, we have

$$\eta(\boldsymbol{\pi}_{new}) \geq \zeta_{\boldsymbol{\pi}_{old}}(\boldsymbol{\pi}_{new}) + \eta(\boldsymbol{\pi}_{old}) - C \cdot D_{KL}^{max}(\boldsymbol{\pi}_{old}||\boldsymbol{\pi}_{new}) \tag{33}$$

The KL divergence has the following property Su & Lu (2022):

$$D_{KL}^{max}(\boldsymbol{\pi}_{old}||\boldsymbol{\pi}_{new}) \leq \sum_i D_{KL}^{max}(\pi_{old}^{ii}||\pi_{new}^{ii}) \tag{34}$$

Based on Eq. 33 and Eq. 34, we have

$$\eta(\boldsymbol{\pi}_{new}) \geq \zeta_{\boldsymbol{\pi}_{old}}(\boldsymbol{\pi}_{new}) + \eta(\boldsymbol{\pi}_{old}) - C \cdot \sum_i D_{KL}^{max}(\pi_{old}^{ii}||\pi_{new}^{ii}) \tag{35}$$

Using Theorem 2, $\zeta_{\boldsymbol{\pi}_{old}}(\tilde{\boldsymbol{\Pi}}_{new})$ and $\zeta_{\boldsymbol{\pi}_{old}}(\boldsymbol{\pi}_{new})$ satisfy the following inequality:

$$
\begin{aligned}
& \zeta_{\boldsymbol{\pi}_{old}}(\boldsymbol{\pi}_{new}) \\
& \geq \zeta_{\boldsymbol{\pi}_{old}}(\tilde{\boldsymbol{\Pi}}_{new}) - \sum_i \frac{1}{2} \max_{s,\boldsymbol{a}} |A_i^{\boldsymbol{\pi}_{old}}(s,\boldsymbol{a})| \cdot \sum_{s,\boldsymbol{a}} \max_s d^{\boldsymbol{\pi}_{old}}(s)^2 + (\tilde{\boldsymbol{\pi}}_{new}^i(\boldsymbol{a}|s) - \boldsymbol{\pi}_{new}(\boldsymbol{a}|s))^2.
\end{aligned}
\tag{36}
$$

By replacing $\zeta_{\boldsymbol{\pi}_{old}}(\boldsymbol{\pi}_{new})$ of Eq. 35 with the RHS of Eq. 36, we have Eq. 32, and thus Theorem 3 is proved.

## B   ENVIRONMENTS

**Exchange (Exc.).** We introduce a new discrete domain setup that delves into the intricate dynamics of agent interactions amidst conflicting interests. Here, $n$ agents are paired with $n$ boxes. Each box contains food for a specific agent but can only be opened by another agent at a cost of 5 reward points. The food's intended recipient gains 10 reward points. Every agent loses 0.01 reward point per time step taken. In a purely self-interested setup, agents would avoid opening boxes for others, leading to a free-rider problem. An illustration of this environment is depicted in Appendix. B, Figure 2(a). The state includes agents' and food items' positions, with random initialization each episode. Agents can move up, down, left, or right, or opt to open a box or stay, based on their alignment with a box.

**Cooperative navigation (Navi.).** Building on the framework from Zhang et al. (2018b), this task requires every agent to approach a landmark. Adopting the same observation and action configurations as in Zhang et al. (2018b), agents earn rewards relative to their proximity to targets. However, they incur a -1 penalty upon collisions. Agents are restricted to exchanging information only with adjacent counterparts. Figure 2(b) in Appendix. B visualizes the environment setup.

**Cooperative predation (Pred.).** We introduce a novel continuous domain task. This environment involves multiple predator agents aiming to capture a single prey, presenting a cooperate-versus-defect conundrum. If all predators cooperate, they each gain a reward of $-1$. In contrast, universal defection results in a $-3$ reward for every predator. A mixed scenario, with only a subset actively engaging the prey, penalizes those participating with a $-4$ reward, while the non-participating predators gain a 0 reward. The challenge is to nurture a cohort of agents favoring cooperative behaviors over

defection-driven individualistic tendencies. For each episode, the prey's position, $x_{tar} \in \mathcal{X}$, and the starting positions of agents, $x_{ag_i} \in \mathcal{X}$, are randomized, with $\mathcal{X} = [0, 30]$. The state is represented as $s^t = [x^t_{ag_1} - x_{tar}, \ldots, x^t_{ag_N} - x_{tar}]$, a continuous variable. Actions fall within $\mathcal{A} = \{-1, +1\}$, corresponding to left and right movements. Figure 2(c) in Appendix. B illustrates this environment.

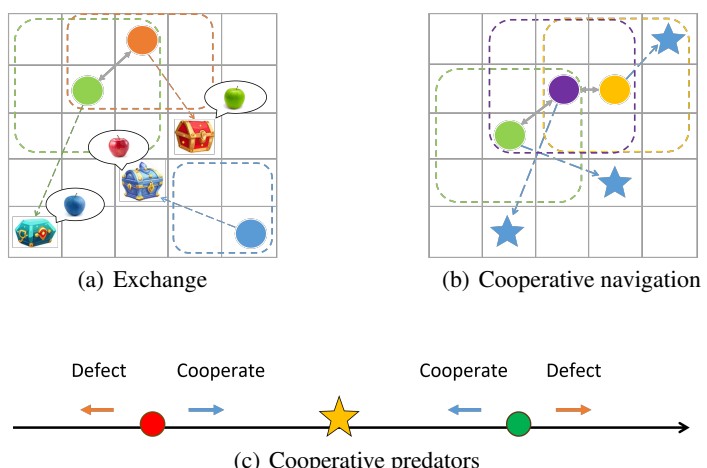

(a) Exchange          (b) Cooperative navigation

(c) Cooperative predators

Figure 2: Illustrations of three environments

## C BASELINES

We compare our AS algorithm to three typical decentralized MARL algorithms that share information under individual reward conditions: value function parameter sharing Zhang et al. (2018b), value sharing Du et al. (2022), and policy parameter sharing Zhang & Zavlanos (2019). For notation simplicity, we refer to these as VPS, VS, and PS respectively. We demonstrate AS's competitiveness despite not relying on value or policy sharing.

VPS employs consensus updates of individual value functions. The updates utilize each agent's unique reward, along with a weighted aggregation of neighboring agents' value function parameters. In contrast, in VS, each agent learns a value function independently without consensus. However, they update individual policy networks based on the average neighborhood value function. PS uses consensus updates to learn global policies. Each agent learns a global policy and aggregates policy parameters from neighbors, but value functions are learned independently without consensus.

## D ADDITIONAL RESULTS

**Assessing impact of neighbourhood range.** To assess the potential impact of the neighborhood range, we vary the maximum neighborhood distance. In the discrete domain, the maximum neighborhood distance is defined as $d = 1$ and 2 units from the agent. In the earlier experiments, $d = 1.5$. In the continuous domain, we set the normalized distance limit as $d = 0.03$ and 0.3, which is 0.1 previously. The results shown in Figure 3 indicate that AS exhibits good performance compared with baselines. In contrast, VPS exhibits a notable performance degradation when the range is decreased in Pred. tasks. VS also degrades in performance slightly with a reduced range. The performance of PS does not improve with an increased range.

**Assessing sensitivity to penalty weight.** We investigate the sensitivity of our algorithm to the weight of the penalty terms in Eq. 15. As an illustrative example, in cooperative predation tasks, the weight is set as $\rho = 0.1$ in the previous experiments. In this part, we set $\rho = 0.03$ and 0.3. Results are shown in Figure 4. The training curves show that our algorithm is robust to the penalty weight in a large scale. The results also confirm that this robustness persists across different agent quantities. On the other side, when using a small weight, the performance exhibits a minor decline. It indicates the importance of the penalty terms in adjusting individual policies towards enhancing cooperation.

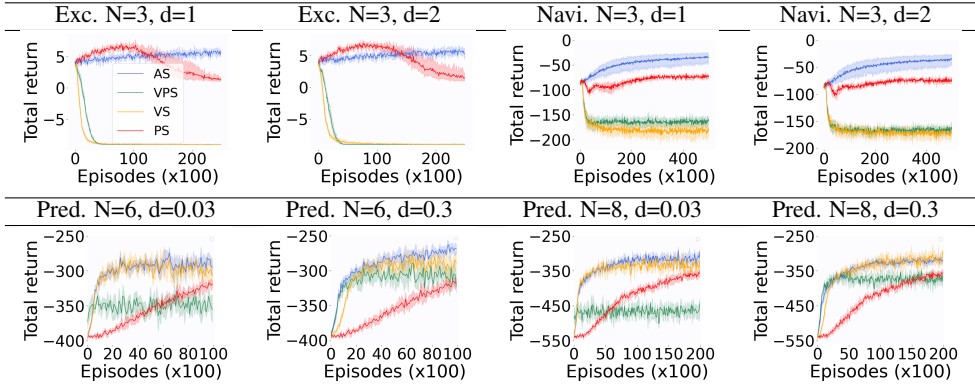

Figure 3: Training curves with different neighbourhood ranges.

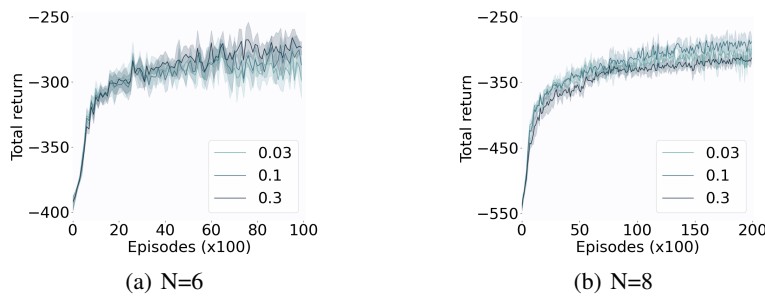

Figure 4: Training curves with different penalty weights.

**Scalability and communication study.** We further consider the scalability of our proposed algorithm when the number of agents increases. A sparse network topology of the decentralized system, where each agent randomly selects another to exchange information, as well as low communication frequency, can help reduce computational complexity. For this study, we tested two different versions of our AS algorithm: (1) each agent randomly selects another agent to exchange information, and (2) the system adopts a random skip (0.5 probability) of communication between steps. When communication is skipped, each agent does not update the anticipated policies of others, and does not use the constraints that constrain the discrepancies between the anticipated action distributions $\pi_{-i}(a_{-i}|\theta_{-ii})$ and the true action distributions $\pi_{-i}(a_{-i}|\theta_{-i})$, i.e. the last two terms in Eq. 15 are removed. Additionally, the policy ratio $\xi_{\mathcal{N}_i}$ involving others' true policies are also removed from Eq. 15. As a result, each agent updates its own policy independently. The same approach is used in the baseline algorithms, i.e. when skipping communication, agents update policies independently. The experimental results using 20 agents in Predation task (Figure 5) demonstrate the effectiveness of our proposed AS algorithm. VS and VPS fall into suboptimal results, while PS cannot learn within the same amount of training episodes.

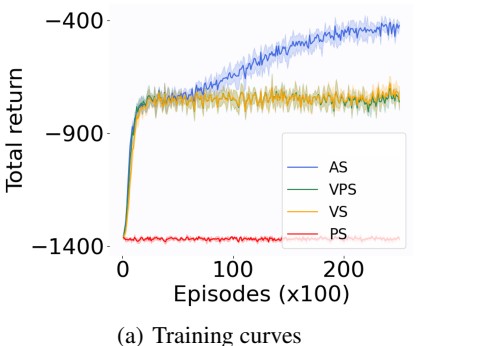 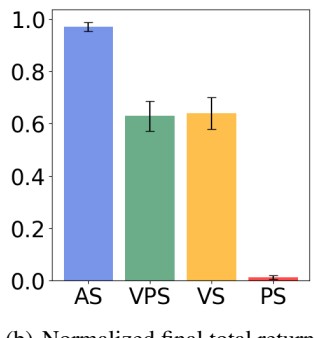

(a) Training curves        (b) Normalized final total return

Figure 5: Experimental results of N=20.

# E ILLUSTRATION FIGURE

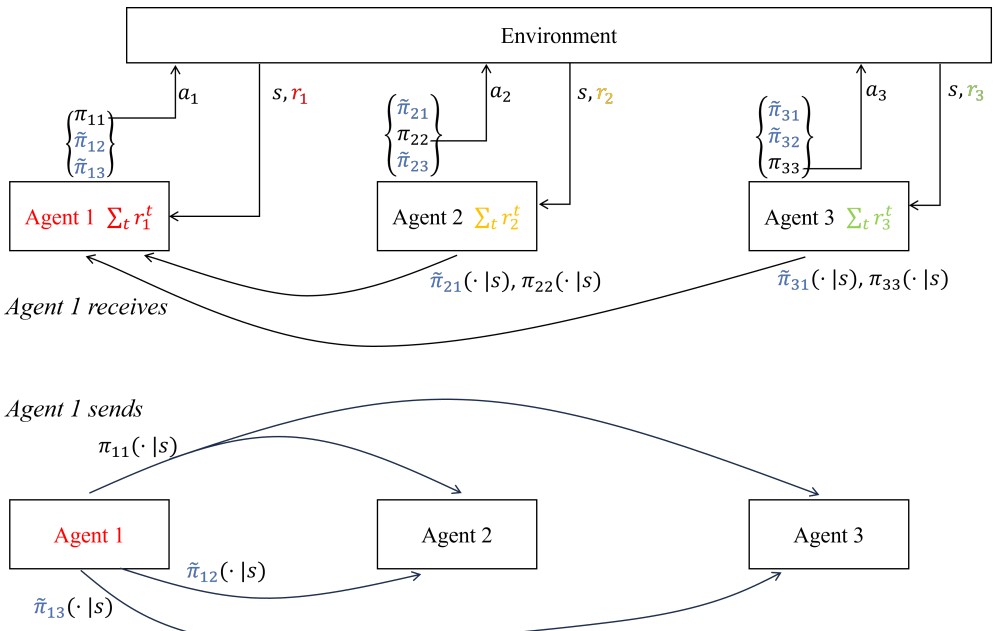

Figure 6: Illustration of AS algorithm, where $d$ represents the function regarding the discrepancy term used in Eq. 15.

# F ALGORITHM

---

**Algorithm 1:** Decentralized MARL via Anticipation Sharing (AS)

---

1   **Initialize**: Policy networks $\tilde{\boldsymbol{\pi}}^i = (\pi^{i1}, \cdots, \pi^{iN})$, value networks $V_i, \forall i \in \{1, \cdots, N\}$ ;

2   **for** *episode = 1 to E* **do**

3      $\mathcal{D}_i \leftarrow \phi, \forall i$ ;

4      Observe initial state $s_1$ ;

5      **for** *t=1 to T* **do**

6          Execute action $a_t^i \in \pi^{ii}(\cdot|s_t)$ ;

7          Observe reward $r_t^i$ and next state $s_{t+1}$ ;

8          Store $(s_t, a_t^i, r_t^i, s_{t+1}) \in \mathcal{D}_i$

9      **end**

10     **for** *iteration = 1 to K* **do**

11        **for** *i=1 to N* **do**

12           Compute advantage estimates $\hat{A}_i^1, \cdots, \hat{A}_i^T$ using Eq 16;

13           Share action distributions $[\pi_{old}^{ii}(\cdot|s_1), \cdots, \pi_{old}^{ii}(\cdot|s_T)]$ to neighbors $\{k \in \mathcal{N}_i\}$ ;

14           **Share anticipated action distributions** $[\pi_{old}^{ij}(\cdot|s_1), \cdots, \pi_{old}^{ij}(\cdot|s_T)]$ to each neighbor agent $j, j \in \mathcal{N}_i$ ;

15           Update $V_i$ using Eq 17 ;

16           Update $\tilde{\boldsymbol{\pi}}^i$ using Eq 15 ;

17           $\tilde{\boldsymbol{\pi}}_{old}^i \leftarrow \tilde{\boldsymbol{\pi}}^i$

18        **end**

19     **end**

20 **end**

---

# G HYPERPARAMETERS

Table 1: Common hyperparameters used in all tasks.

| Hyperparameter | Value |
|---|---|
| Critic network | (128, 64, 1) |
| Actor network | (128, 64, dim_a) |
| Activation | ReLU |
| Optimizer | Adam |
| Critic learning rate | 1e-4 |
| Discount factor $\gamma$ | 0.99 |
| GAE $\lambda$ | 0.98 |
| Clipping $\epsilon$ | 0.2 |
| Update iteration $K$ | 3 |

Table 2: Different hyperparameters used for discrete and continuous domains.

| Domain | Continuous | Discrete |
|---|---|---|
| Actor learning rate | 1e-4 | 5e-5 |
| $\rho$ | 0.1 | 1e4 |

