# OpenReview forum: "DECENTRALIZED MULTI-AGENT REINFORCEMENT LEARNING VIA ANTICIPATION SHARING"
_ICLR.cc/2024/Conference — Submitted to ICLR 2024_

### Official Review · Reviewer_f5qP · 2023-10-31

**Soundness:** 3 good
**Presentation:** 2 fair
**Contribution:** 3 good
**Rating:** 6
**Confidence:** 3

**Summary:**

This paper addresses the mixed motive problem setting, where each individual agent only has access their own individual rewards, but the system objective is to maximize social welfare (defined as the sum of individual agents' returns). The authors propose a decentralized learning method called Anticipation Sharing (AS) based on teammate modelling. The key idea is that each agent j will predict their teammates' actions (teammate modelling), and maximize a surrogate objective based on the teammate models, subject to a KL constraint. At each update step, each agent shares both their own actions and the predicted teammate actions with neighboring teammates. In the proposed method, rewards, values and model parameters are not shared. The empirical evaluation is performed on three mixed-motive tasks: exchange, cooperative navigation, and cooperative predation, and shows that (1) AS achieves higher returns than baseline methods, (2) AS is robust to number of neighbors / neighbor distance hyperparameters

**Strengths:**

- The paper is addressing an understudied and important problem, that of the mixed motive setting where agents must cooperate and compete with others to achieve their own objectives. Based on the strength of the empirical results, the contribution is good.
- The mathematical notation is mostly well-defined, and the proofs are written rigorously. I did not discover any major mistakes or inconsistencies in the paper, and the authors have described their theory, algorithm, and experiments in enough detail for others to replicate.
- The empirical results seem strong. The proposed method seems to converge more stably and to a higher return than related methods. The experiments figure is also nicely done (figure text size is appropriate, colors are well differentiated, and legend is fixed across figures for easy comparison).

**Weaknesses:**

The biggest weakness of this paper is that it fails to establish why AS has better assumptions/performs better than related methods. It also fails to provide intuition for the method.
- While the authors have provided many details on AS and have good results, fundamentally, the authors failed to communicate to me answers to the following important questions. What is the motivation for AS? Put another way, what are the existing methods and why would we expect AS to do better than them? Put a third way, what is the problem with the existing methods that AS solves?
- The abstract states, "In this research, we obviate the need for sharing rewards, values, or model parameters." Sharing action distributions is not a weaker assumption, since it involves sharing a similar amount of information. So why is it a better assumption?
- The intuition behind the proposed method and the motivation isn't totally clear to me. In Definition 2, the authors define the expectation over individual anticipated advantages.  If I understood correctly, this is the ith agent's advantage of the modelled joint action over the joint actions sampled from π^′, which I believe is the value of the true joint policy (but am not totally sure). As each agent has a different reward function, the advantages may not even be on the same scale. Thus,  it seems dubious to average over all agents' advantages.  Can the authors provide further intuition for what this quantity means, and why we might wish to bound the difference

The second biggest weakness of the paper is the experimental analysis.
- The current analysis is too speculative and makes vague statements. For example, the authors have the following statement in the experimental section:

	"In comparison, the performance of VS and VPS is not stable across all the tasks. This may imply that the mere act of sharing value or value functions and achieving a consensus on values may not be sufficient to establish cooperative policies. This is because, even though value updates can approximate a system-wide value, **the policy updates of agents lack a coordination mechanism, thereby leading to inferior cooperation performance.** Our algorithm provides a policy coordination mechanism, which directly rectifies the deviation of individual policy update from the collective cooperation goal."

The bolded claim is a causal statement that is unsubstantiated by evidence. It is simply a speculation by the authors. The authors should either provide an experiment to support this claim, OR clearly mark it as a hypothesis. Overall, this section of the experimental analysis provides only weak insight into why AS is better than the baselines.

- Missing experiment: It seems a bit impractical for each agent to have an accurate anticipated policy for all other agents. It would limit the scalability of the method. Can the authors add an experiment on, and discussion of how inaccuracies in teammate modelling affect the performance of the method?

The following issues are relatively minor, but I would still like to see them corrected.
1.  Miscellaneous clarity issues:
- There should be a nice figure summarizing the method, somewhere in the paper.
- The language of 'anticipated policies' isn't clear. It would be much clearer (and more reflective of what the paper is doing) if the paper replaced the terminology "anticipation" by "teammate modelling".
2. Algorithm 1, lines 13 and 14: this notation seems that the policies are shared, not the action distribution. Perhaps the authors can add a clarifying note to the caption.
3. The paper makes overbearing claims. For example, in the abstract, the authors state their method represents a "paradigm shift". I feel that the results don't support such a strong claim.
4. Missing related work: there is a large body of related work on teammate modelling, both in the CTDE setting and the ad-hoc teamwork setting. Please see Albrecht et al.'s survey (https://arxiv.org/abs/1709.08071), and add an appropriate subsection of related work.

**Questions:**

See the weaknesses section.

---

> ### Author Response · Authors · 2023-11-17
> **Response to Reviewer f5qP (part 1)**
>
> Thank you for your positive feedback on our paper. We appreciate your recognition of the clarity and rigor in our explanations and proofs, and we're pleased that our figures effectively showcase our research findings.
>
> **Motivation for AS and comparison with existing methods**. Thanks for pointing out that the proposed methodology has not been motivated sufficiently. We have taken this important observation into account in our revised manuscript.
>
> In summary, the primary motivation for developing AS was to address the limitations of existing decentralized MARL methods, which often rely on sharing rewards, values, or model parameters. These conventional methods can lead to challenges such as privacy concerns and computational inefficiencies. AS offers a novel approach by focusing on sharing action distributions, which we argue is potentially less invasive and more privacy-preserving for agents compared to exposing detailed model parameters or reward functions directly. This shift in information sharing paradigm is central to AS's proposed contribution. We hypothesize that by sharing anticipated actions instead of full internal model details, agents may be able to achieve more effective coordination while maintaining a greater degree of autonomy and privacy.
>
> We also wish to emphasise that anticipation sharing was undertaken in a distributed manner during training only. The resulting learned policy networks of other agents are not activated at test time. In the revised manuscript (Appendix.E, page 17), we have included a system diagram to help explain the distributed, private nature of the AS approach visually.
>
> **Why sharing action distributions is a better assumption**. Sharing action distributions, as opposed to rewards, values, or model parameters, strikes a balance between the need for effective coordination and maintaining individual agents' privacy. While it does involve sharing a significant amount of information, it circumvents the direct exposure of an agent's internal decision-making process, which can be sensitive in many applications. We believe that this approach can be particularly advantageous in scenarios where privacy and security are paramount, and where the sharing of detailed internal states or policies is not feasible. We have stressed this point more in the revised version.
>
> **Definition 2.**  We appreciate your attention to Definition 2 and the opportunity to clarify it further given its importance. It appears there may have been a misunderstanding regarding the expectation over individual anticipated advantages. This expectation is indeed calculated over the anticipated joint actions, denoted as $\tilde{\pi}^i$, not over $\pi'$. The key distinction here is that each agent in our framework optimizes based on its anticipated joint policy ($\tilde{\pi}^i$), rather than $\pi'$. This is a crucial aspect of our approach, as it enables decentralized optimization - each agent individually anticipates and adapts to the actions of others without the need for central coordination or access to the true joint policy. This decentralized nature of policy anticipation and adaptation is what allows our method to be applicable in settings where sharing detailed internal models or policies is not feasible. In the revised manuscript, we have elucidated this point more clearly to avoid any confusion.

---

> > ### Author Response · Authors · 2023-11-17
> > **Response to Reviewer f5qP (part 2)**
> >
> > **Experimental analysis.**  Thank you for the insightful feedback on strengthening our experimental analysis section. We appreciate you highlighting areas needing additional empirical substantiation and clarity.
> >
> > To address these concerns,  in the revised manuscript (Section 5.2) , we have included a more comprehensive analysis of the experemental results. We have also clearly denoted certain statements as hypotheses where they remain partially speculative.
> >
> > Importantly, we want to clarify that our method does not focus on explicitly modeling other agents' policies. Rather, each agent calculates anticipations about others' policies to optimize its own objectives. In particular, the anticipation $\pi^{ij}$ is solved by agent i to optimize its own objective (Eq. 11). The constraints in Eq. 11 incorporate the discrepancy between $\pi^{ij}$ and agent j's true action distribution $\pi^{jj}$. While related to teammate modeling, the key distinction is that this discrepancy is not the primary optimization target - it is a constraint when maximizing agent i's objective.
> >
> > $\pi^{ij}$ is then shared with agent j so it can incorporate agent i's objective when maximizing its own. Additionally, agent i's policy $\pi^{ii}$ is adjusted based on the discrepancy with $\pi^{ji}$, agent j's anticipation of agent i. This enforces agreement between local learning and others' anticipations, going beyond typical teammate modeling objectives.
> >
> > Regarding the scalability of the method, in the revised Appendix of the manuscript (page 16), we include experiments regarding scalability study.  Please let us know if the revised experiments help assuage the concerns over scalability.
> >
> > **Miscellaneous clarity issues.** To improve the overall clarity, as per your suggestion,  in the revised manuscript (Appendix.E, page17), we have incorporated a comprehensive figure summarizing our method, which should aid in better understanding the overall approach- thanks for this.
> >
> > **Relation to teamate modelling.** We appreciate your suggestion regarding the clarity of the term 'anticipated policies' in our paper. However, after careful consideration, we have chosen to retain the term "anticipation" rather than adopting "teammate modeling." This decision is based on the fundamental differences between these concepts. Unlike traditional teammate modeling, which involves agents modeling or predicting the actual policies of their peers, our concept of "anticipation" refers to each agent calculating actions of others that would be most beneficial for its own objective. Therefore, the anticipations include the information about individual objectives implicitly. By exchanging the anticipations, individual agents can balance others' objectives and thus the collective performance when optimizing its own objective.
> >
> > This approach emphasizes solving anticipations serving for the agent's own return optimization, and strategic adaptation based on the anticipations from others that implicitly include the information of their returns, rather than attempting to accurately model their behaviors. This distinction is central to our approach and sets it apart from conventional teammate modeling. In the revised manuscript, we have clarified this terminology and its importance to our research, highlighting how our method's unique perspective contributes to the field of decentralized multi-agent environments. Please see the last paragraph in Section.2 and the last paragraph in Section 4.2 in the revised manuscript.
> >
> > **Clarification in algorithm 1.** Thanks for pointing out this clarity issue. We have revised lines 13 and 14 of Algorithm 1 to clearly indicate that it is the action distribution, rather than the policies themselves, that are shared.
> >
> > **Missing related work.** We appreciate the reference to Albrecht et al.'s survey on teammate modeling. We have added teammate modeling in Related Work (Section. 2) in our revised manuscript.

---

> > ### Comment · Reviewer_f5qP · 2023-11-21
> >
> > Thank you for your revised manuscript with the emphasis on privacy, the section on teammate modelling, and the new Figure 6. The new abstract and intro is clearer, but I encourage the authors to update the abstract to explicitly state that their paper addresses a wider set  of cooperative problems than the commonly studied, fully cooperative setting.
> >
> > The new Figure 6 cleared up a major point which I previously misunderstood --- namely, that agent $i$ shares its anticipation of agent $j$ *with* agent $j$, rather than sharing its own action distribution with agent $j$. I highly encourage the authors to add this figure to the main paper, as it helps the reader immediately understand the key points of the main paper. Due to this understanding, coupled with the new section on teammate modelling,  I'm okay with the authors keeping the terminology of "anticipation sharing",  rather than teammate modelling
> >
> > I will raise my score.

---

> > > ### Author Response · Authors · 2023-11-22
> > > **Response to Reviewer f5qP**
> > >
> > > Thank you for your comments. We appreciate that you spent time reading our revised manuscript and replies. We’re pleased that you understood our method and would like to thank again for your initial suggestions. We also sincerely thank you for your valuable suggestions provided above. Exactly, we address a wider and more challenging set of cooperative problems where agents only receive individual rewards instead of team rewards, and cooperate to maximize the total return of all the agents without sharing rewards or values or network parameters. We will further emphasize this point and add Figure 6 to main paper in our manuscript.

---

### Official Review · Reviewer_NPvi · 2023-10-31

**Soundness:** 3 good
**Presentation:** 4 excellent
**Contribution:** 3 good
**Rating:** 6
**Confidence:** 4

**Summary:**

In decentralized multi-agent reinforcement learning (MARL), aligning individual and collective goals is challenging. Traditional methods share rewards or models, risking discoordination and privacy breaches. This study introduces a new approach that bridges this gap without sharing parameters, emphasizing anticipation of other agents' actions. A novel MARL method with anticipation sharing is presented.

**Strengths:**

1. The article presents "Anticipation Sharing", a pioneering decentralized MARL method. This method enables agents to update anticipations about neighboring agents' action distributions, offering a fresh perspective in MARL.
2. Both theoretical analysis and simulated environment tests have been employed to validate the proposed approach, ensuring its robustness and applicability in diverse settings.
3. By addressing prevailing challenges like policy discoordination and privacy concerns, the research aptly underscores its relevance and timeliness to the current state of MARL.
4. Real-world implications, including the complexities of decentralized learning and agents' privacy preferences, are thoroughly discussed, highlighting the practical significance of the study.

**Weaknesses:**

1. The simulated environment tests presented seem overly simplistic. Without testing in more complex scenarios, the method's scalability and robustness remain unproven.
2. The paper's narrative occasionally lapses into redundancy, indicating a need for more concise and focused writing.

**Questions:**

1. How does the proposed "Anticipation Sharing" method handle non-stationarity introduced by agents continuously updating their anticipations? Continuous updates can lead to a moving target problem in MARL.
2. While the paper addresses the challenges of policy discoordination and privacy concerns, how does the proposed method handle issues related to credit assignment, especially when agents have conflicting goals or when their contributions to the global reward are imbalanced?
3. The simulated environment tests mentioned in the paper appear to be somewhat simplistic. Were any tests conducted in environments with high agent heterogeneity or with more complex interaction dynamics? If not, how can the effectiveness of the method be ascertained in such scenarios?

**Details Of Ethics Concerns:**

Looks fine.

---

> ### Author Response · Authors · 2023-11-17
> **Response to Reviewer NPvi**
>
> We are pleased that our AS approach in decentralized MARL is recognized as innovative. We are also glad that our approach of combining theoretical analysis with simulated environment tests has been acknowledged.
>
> **Scalability and robustness.** Our simulated tests in relatively simple environments are intended as preliminary validations of the fundamental approach. We recognize that demonstrating scalability and robustness requires evaluation across more heterogeneous, dynamic scenarios.
>
> Per your insightful guidance, in the revised Appendix of the manuscript (page 16), we have expanded the manuscript to include experiments regarding scalability study.  Please let us know if the revised experiments help assuage the concerns over scalability and robustness testing.
>
> **Redundancy in writing**. We have taken your feedback about the occasional redundancy in our writing into consideration. The revised manuscript has been edited accordingly.
>
> **Handling non-stationarity**. Our method addresses this challenge by incorporating a mechanism designed to predict and adapt to changes in agents' policies over time. This approach is proactive in nature, aiming to stabilize the learning process and enhance coordination among agents by reducing the unpredictability that arises from these continuous updates. By anticipating potential policy shifts of other agents, each agent can adjust its strategy more effectively, leading to a more stable learning environment. However, fully addressing the complexities of non-stationarity in MARL remains an open challenge requiring further research. This will be a focus as we continue developing the anticipation sharing approach.
>
> **Credit assignment in conflicting goals**. Credit assignment in multi-agent reinforcement learning typically arises in settings where a singular team reward is shared among agents. Our research, however, focuses on decentralized learning environments where such a team reward is not available, and each agent only has knowledge of its individual reward. In these scenarios, rewards may often be in conflict, meaning an increase in one agent's reward could potentially lead to a decrease in another's.
>
> In our setting, the challenge is not so much about assigning credit from a collective reward, but about maximizing individual returns in a way that also contributes to the overall system's effectiveness. This is where our anticipation sharing framework plays a crucial role. It aims to improve coordination among agents by enabling them to balance their own objective and other agents’ objectives via constraining the policy with the anticipations from others, since the anticipations from others contribute to maximizing other agents’ objectives.
>
> While our approach focuses on enhancing coordination to potentially reach a maximal total return, it does not directly solve the credit assignment problem typical in environments with shared team rewards. Instead, it addresses the unique challenges of decentralized learning where independent reward maximization might lead to a Nash Equilibrium, rather than a collective optimum. This distinction is critical to understanding the context and objectives of our research in decentralized multi-agent systems.
>
> We have edited the manuscript to make these points clearer, such as the second paragraph of the Introduction, and the two paragraphs after Eq.11 in Section 4.2.

---

### Official Review · Reviewer_FSUM · 2023-10-31

**Soundness:** 1 poor
**Presentation:** 1 poor
**Contribution:** 1 poor
**Rating:** 3
**Confidence:** 4

**Summary:**

The paper is about a solution for multi-agent reinforcement learning. The solution refers to it as "anticipation sharing". In my opinion, the anticipation the authors refer to is essentially an estimation of the values of the other agents. The authors provide a series of "theorems", which are presented as an extension of results in the original TRPO paper. The validation is based on a series of experiments value function parameter sharing, value sharing and policy parameters sharing.

**Strengths:**

+ The problem of decentralized RL has been widely explored, but, in actual truth, it can be considered still a rather open problem.

**Weaknesses:**

- It is unfortunately quite difficult to understand the actual mathematical underpinnings of the paper. The first part presents a series of "theorems". However, their proofs do not appear convincing. The authors do not prove the inequalities reported in the Section 4.1 (more specifically, Theorem 1, Theorem 2 and Theorem 3). Then the authors introduce a surrogate optimization objective (in Section 4.3). However, the authors do not justify the introduction of this surrogate objective. It is very difficult for the reader to understand why it is a good idea such a function, which is quite complex computationally.
- The reviewer struggles to understand the introduction of the constraints in Section 4.3. This might be due to a presentation issue, but the derivations of the constraints from the optimization problem listed in Section 4.2 is not straightforward.
- The results reported in the evaluation are difficult to explain. Why do some of the methods perform so badly?
- For the Pred. task the performance appear very similar to value sharing.

**Questions:**

- What is the difference between value estimation of the other agents vs anticipation?
- What is the advantage in using the surrogate optimization objective in Section 4.3?
- Can you please motivate/derive the constraints in Section 4.4?
- Can you please explain the results in Figure 1?
- Can you please explain why we observe difference in terms of performance for the various tasks? Why does your method perform better for certain environments? If the difference is low, it seems that value sharing would be a preferable solution given its lower complexity. What is the trade-off a practitioner should consider in this situation?
- Can you please explain this sentence: "Theoretically, we established that the difference between agents’ actual action distributions and the anticipations from others bounds the difference between individual and collective objectives."? I am not sure that is the actual theoretical result that you showed in the first part of the paper. Alternatively, this might be considered as an expected fact, which does not need a proof?

---

> ### Author Response · Authors · 2023-11-17
> **Response to Reviewer FSUM (part 1)**
>
> **Response to Weakness:**
>
> **Mathematical underpinnings and proofs.** Thank you for your careful reading of our paper and for raising these important points. We appreciate you taking the time to provide such thoughtful feedback.
>
> Regarding Section 4.1, we want to clarify that the key inequalities were already defined and proved in the original Appendix. We have aimed to improve the presentation and clarity of the corresponding theorems in the main text, to better highlight their role in building our proposed solution and strengthen the logical flow. The detailed proofs remain in the Appendix. Please let us know if the revised framing of the theorems makes their implications and connectivity clearer, given that the foundational inequalities were established in the first submission.
>
> We also appreciate you noting the need for more background on the surrogate optimization objective introduced in Section 4.2. In our revised manuscript, we have worked to motivate and explain this component more completely, in order to elucidate its rationale and benefits.
>
> **Surrogate Objectives and Constraints in Section 4.2 and 4.3.** Thank you for your suggestions regarding the presentation of the surrogate objective and the constraints in Section 4.3. Upon reflection, we agree the original drafting could be improved for clarity and accessibility.
>
> In Section 4.2, we leverage the lower bound given by Theorem 3 to maximize the orginal collective return, since the collective return is intractable without a global view. By omitting irrelevant terms, we have a surrogate objective Eq. 10. Further, to make it feasible in decentralized MARL, we reformulate it from each agent's limited perspective. Remarkably, we can distill the relevant components into a local objective and constraints for each individual agent, which leads to Eq.11. To practically solve Eq. 11, in Section 4.3, we propose an algorithm, involving specific steps, each targeting different aspects of the optimization challenge. In the revised manuscript, we have rewritten Sections 4.2 and 4.3 to explain the surrogate objective and the constraints in a more intuitive, step-by-step manner. Please let us know if the revised explanation provides adequate insight into how the surrogate objective and the constraints emerge from the optimization problem. We welcome any additional feedback on making this section clear and complete.
>
> **Evaluation results**: In our revised manuscript (Section 5.2), we delve into a more detailed analysis to explain the varying performance levels of different methods, including our own.
>
> However, it is important to note that the primary aim of our study is not to outperform the baseline algorithms in every scenario but to provide a viable alternative in more complex and heterogeneous environments. Our approach is especially relevant in settings where agents cannot exchange values or rewards due to privacy constraints. In such scenarios, our method offers a means of coordination without compromising individual agent privacy.
>
> Additionally, our method is underpinned by a theoretical framework designed for stability across a variety of tasks. This theoretical basis may contribute to more consistent performance, as opposed to methods like VS, which might excel in certain tasks but struggle in others.
>
> **Performance in the Predation task**:  Thank you for raising this insightful point regarding the similar performance observed in the Predation (Pred.) task compared to value sharing (VS). We acknowledge that this outcome warrants further investigation in our revisions.
>
> Our initial hypothesis is that the characteristics of the Pred. task, where agents are homogeneous with a common target, might inherently lend themselves to approaches like VS, thereby explaining the similarity in performance.
>
> In the revised manuscript (Section 5.2) , we have included a more comprehensive analysis of the performance in the Pred. task, exploring why our approach shows similar effectiveness to VS in this particular scenario and highlighting the distinct advantages our method offers in diverse and privacy-sensitive environments.

---

> ### Author Response · Authors · 2023-11-17
> **Response to Reviewer FSUM (part 2)**
>
> **Response to Questions:**
>
> **Difference between value estimation and anticipation**: Thank you for giving us the opportunity to clarify the important difference between value estimation and anticipation.
>
> Calculating the value function of other agents involves estimating the expected returns or outcomes of their actions, typically based on observed behaviors and strategies. The focus is on understanding what outcomes other agents aim to achieve.
>
> In contrast, as defined in our work, anticipation goes beyond mere outcome estimation. The anticipation $\pi^{ij}$ is solved to optimize agent i's objective (Eq. 11). $\pi^{ij}$ is then sent to agent j, so agent j can balance agent i's objective when maximizing its own. Additionally, agent i's policy $\pi^{ii}$ is adjusted based on the discrepancy with $\pi^{ji}$, agent j's anticipation of agent i. This constraint further enforces agreement between local learning and others' anticipations.
>
> Therefore, the anticipations include the information about individual objectives implicitly. By exchanging the anticipations, individual agents can balance others' objectives and thus the collective performance when optimizing its own objective. We highlighted this point in our revisded manuscript, see the two paragraphs after Eq.11 in Section 4.2.
>
> We hope that the revised manuscript does a better job at highlighting this conceptual nuance.
>
> **Advantage of using the surrogate optimization objective**: In the original objective (Eq.1), the optimization target is the true joint policy, which is infeasible in decentralized settings where each agent does not have access to other agents' true policies and cannot dictate to others how to update their policies.
>
> The surrogate objective stems from the lower bound (Eq.9) of the original objective, which can be decomposed into individual objectives that optimize the anticipated joint policy consisting of the agent’s own policy and policy anticipations to other agents. The anticipated joint policy is solved by each agent, which is not the true joint policy composed of the true individual policy of each agent, as defined in Definition 1.
>
> The surrogate optimization objective introduced in Section 4.2 offers a more tractable solution to the complex problem of multi-agent coordination under uncertainty. It simplifies the optimization landscape, making it feasible to find solutions that, while not globally optimal, are practically effective. This objective also helps in balancing the computational complexity with the benefits of improved coordination, which is particularly valuable in environments where exact solutions are computationally prohibitive.
>
> We believe that the revised manuscript (Section 4.2) is significantly clearer about these key aspects.
>
> **Motivation and derivation of constraints in Section 4.2**:  Thank you for allowing us to clarify the origins of the constraints presented in Section 4.2 (Eq.10 and Eq.11). We appreciate you highlighting this component for further discussion.
>
> You raise an excellent point - the constraints are directly motivated by the lower bound in Section 4.1, Eq. 9. Specifically, the constraints in Eq. 10 derive directly from Eq. 9, with the focus on optimizing $\pi_{new}$ while fixing $\pi_{old}$. They ensure the surrogate optimization aligns with multi-agent practicalities.
>
> Further, Eq. 11 stems from separating out the agents in Eq. 10's constraints and anticipations. This separation is key for decentralized optimization where each agent acts based on its policy and anticipations of others.
>
> We have expanded the manuscript to provide a detailed, step-by-step explanation of how these constraints emerge from the underlying optimization formulation. We aim to elucidate their motivations and relevance to the MARL setting.
>
> **Explanation of Results in Figure 1**: Figure 1 illustrates the performance of our method in various simulated environments. We have enhanced this Figure's explanation by detailing the specific characteristics of each environment and how they interact with our method. This includes a discussion on the factors contributing to the observed performance trends, providing a clearer understanding of the effectiveness of our approach in different settings. Please see Section 5.2 in the revised manuscript.

---

> ### Author Response · Authors · 2023-11-17
> **Response to Reviewer FSUM (part 3)**
>
> **Performance for certain environments:** Our approach is especially relevant in settings where agents cannot exchange values or rewards due to privacy constraints. In such scenarios, our method offers a means of coordination without compromising individual agent privacy. Therefore, the aim of our study is not to outperform the baseline algorithms in every scenario but to provide a viable alternative in such challenging settings. Additionally, our method is underpinned by a theoretical framework designed for stability across a variety of tasks. This theoretical basis may contribute to more consistent performance, as opposed to methods like VS, which might excel in certain tasks but struggle in others.
>
> **Clarification of theoretical statement**: You are correct that this statement (”theoretically, we established that the difference between agents”) stems from our theoretical framework, particularly Theorem 3 and Eq. 9 as described in the manuscript. The LHS of Eq.9 represents the original multi-agent collective objective. The first term on the RHS is the summation of individual objectives from Definition 2 and Eq.6.
>
> Critically, the last term in Eq.9 encapsulates the difference between actual action distributions and anticipations from others. Therefore, as you noted, Eq.9 establishes that this difference bounds the discrepancy between individual and collective goals. This highlights the significance of anticipation sharing for aligning behaviors with group objectives.
>
> Per your helpful feedback, we have significantly revised the presentation of the methodology in the new version of the manuscript to address this point more clearly. Please let us know if the updated explanation provides adequate clarification on how this key theoretical result relates agents' anticipations to collective outcomes. We appreciate you highlighting this opportunity to strengthen the clarity of our framework.

---

> > ### Comment · Reviewer_FSUM · 2023-11-20
> > **My Assessment after the Rebuttal of the Authors**
> >
> > I acknowledge that I read the comments of the authors.
> >
> > - I have still clear concerns about the theoretical foundations of the approach. In particular, the introduction of the surrogate functions (and its advantages) are still unclear to me. Thanks for the additional information, but it appears potentially impractical, especially in a cooperative scenario?
> > - After receiving your clarifications, it seems to me that your approach is essentially a semi-centralised one. But then, why not using a fully-centralised one? what are the advantages? I do not see any good reason unfortunately to have something in between as the approach you propose. Fundamentally, I miss the justification of the approach, which appears only as a variation of existing approaches, without a clear compelling motivation.
> > - What is the guarantee that "This constraint further enforces agreement between local learning and others' anticipations." This key point does not seem proven in terms of convergence.
> > - The authors did not provide explanations about the experimental results, which are very difficult to understand and justify.
> >
> > The rebuttal was useful to clarify my understanding of this work, and, consequently, I confirm my assessment of this paper.

---

> > > ### Author Response · Authors · 2023-11-21
> > > **Response to Reviewer FSUM**
> > >
> > > Thank you for your reply.
> > >
> > > **Theoretical foundations.** Could you clarify which specific aspects you found impractical or unclear? We took your initial comments on board and our revised manuscript addressed your initial concerns. We’d happy to expand on any details that need further explanation.
> > >
> > > **Semi-centralised or fully-centralised approach.** We’re not entirely clear about the reviewer’s meaning of semi-centralised learning. To clarify again, there is no single central agent or parameter server in our approach. However, agents do share limited information with neighbours. We have updated the terminology to reflect this more accurately. The key advantage over fully centralised approaches is avoiding the need for a central controller with access to global policies.
> > >
> > > **Existing approaches and our motivation.** In response to your previous comments, in the revised manuscript we uploaded, we have already expanded the Introduction to more clearly articulate the limitations of existing methods and our specific motivations, especially regarding privacy concerns and communication overheads with sharing rewards, values, or model parameters. Please refer to the Introduction, especially “*In real-world applications, the issue of privacy, particularly concerning rewards and values, becomes a significant hurdle. Agents often prioritize keeping this information confidential, posing a challenge to the practicality of methods that require such sharing. Additionally, sharing model parameters incurs substantial communication overhead and also privacy concerns, which can also result in the transfer of excessive and non-essential information, thereby slowing the learning process*”. We believe we have made our motivation clearer now, and have related our approach to existing ones in the Related Work section.
> > >
> > > **Convergence proof.** We acknowledge the lack of a complete convergence proof is a potential limitation. However, providing rigorous guarantees for decentralised multi-agent deep RL remains an open challenge that we aim to address more fully in future work. The policy update clipping and discrepancy penalty are practical mechanisms to improve stability, but do not constitute a proof. Investigating approaches with more robust guarantees is an important direction. The introduced policy update mechanism through clipping, which is similar to PPO, enhances algorithm stability by preventing drastic policy changes during the training process. Empirically, this helps the algorithm achieve convergence across tasks, as demonstrated in the paper. Regarding the “agreement between local learning and others' anticipations”, Step 2 in Section 4.2 is about using the discrepancy penalty when the discrepancy between local learning and others' anticipations is getting larger, and thus enforces agreement between local learning and others' anticipations.
> > >
> > > **Explanations about the experimental results**. You make a fair point that providing definitive explanations of algorithm performance differences is beyond the scope here. As you note, our primary contribution is presenting a practical decentralised method without needing to share rewards or values. For full transparency and reproducibility, our code will be publicly available. This should allow others to more easily replicate the experiments and potentially provide additional insights into the algorithm behaviours. However, fully analysing and explaining the behaviour of every other baseline is challenging and goes beyond the intended scope of this initial work. We have removed any strong claims about performance explanations.

---

### Official Review · Reviewer_yNQQ · 2023-11-01

**Soundness:** 3 good
**Presentation:** 3 good
**Contribution:** 2 fair
**Rating:** 5
**Confidence:** 2

**Summary:**

The manuscript describe a method through which the optimization step of a multi-agent learning problem is constrained with a step discrepancy bound. The authors show the bound requires global policies, which indeed violates the decentralizability constraint of the advertised problem. The paper then propose a more local approach with estimating and sharing of "anticipated policies" as a more practical replacement.

**Strengths:**

The paper is written with generally mathematical definitions, and the proposed clipping method makes intuitive sense, despite requiring a certain leap of faith. Given the popularity of PPO, I have reasons to believe that the method should provide some performance gains.

**Weaknesses:**

The proposed method is still inherently a centralized approach since information regarding policies is, strictly speaking, being shared in the communication network of the networked MMDP. The centralization issue, however, can be ignored if the proposed method can be shown to avoid local nash such as seen in well-known dilemmas. Unfortunately, the authors do not mention optimality of the solutions, and intuitively, I do not think the method will solve the optimality problem. In addition, I think related works of this paper should also include some papers on opponent modeling. Some convergence results are already available in the case of policy prediction.

minor: in proof for theorem 1, after eq.20, you have \E_s^\prime where it should have been \E_{s^\prime}

**Questions:**

what happens in more challenging scenarios where there are actual dilemmas? have you considered partially observable settings? would the theory still sound in partially observable environments?

---

> ### Author Response · Authors · 2023-11-17
> **Response to Reviewer yNQQ (part 1)**
>
> **General comments.** Thank you for your positive remarks on our mathematical approach and the intuitive appeal of our proposed clipping method. We share your optimism that, given the success of PPO in various settings, our method is likely to yield performance improvements in decentralized MARL scenarios.
>
> **Centralisation issue.** In addressing your concerns regarding the centralization aspect of our method, we firmly assert that our approach fundamentally diverges from conventional centralized learning systems. Firstly, each agent in our system operates with autonomy, learning policies based on their own reward signals and shared anticipations, without any central unit aggregating all information. This decentralized framework inherently differs from traditional centralized methods where a single entity controls or processes all information.
>
> Secondly, the sharing of policy information in our model does not equate to full centralization. Agents exchange only their action distributions for the states they encounter, rather than sharing entire policy parameters. This partial information sharing, coupled with the individual reward knowledge each agent possesses, ensures that our method maintains a decentralized essence. The theoretical proofs are based on a fully-connected network topology, which gives a condition of the optimality. Our practical algorithm provides a general setup where the network topology does not need to be fully-connected. Experimental results demonstrate descent performance in several different tasks.
>
> We emphasized these two points in our revised paper. Specifically in the last paragraph in Section 4.2 and the first paragraph in Section 4.3.
>
> **Local Nash equilibria.** This question is critical to understand the problem we solved in this work. When each agent optimizes its own return without considering other agents, they will reach Nash equilibira, which might be local optimum. Social dilemma examplifies this problem. However, the aim of our method is to achieve global optimality (Eq.1) instead of local Nash equilibria. In a decentralized multi-agent system, no single agent has access to global information about the full joint policy or other agents' exact policies. Our method derives a lower bound of the global objective, which can be decomposed into individual objectives that involve the agent’s own policy and policy anticipations to other agents. The anticipations are different from predictions or estimations of other agents’ decisions used in opponent modelling. Specifically, the anticipation of agent i to agent j, i.e., $\pi^{ij}$, is solved to optimize agent i’s objective (Eq.11). By maximizing the lower bound of the global objective, the global optimality rather than local Nash equilibria can be achieved.
>
> Based on the theoretical results, we proposed a practical algorithm to solve Eq.11. Deriving rigorous guarantees for convergence is notoriously challenging in decentralized multi-agent reinforcement learning problems because of this complexity. In future work, we aim to explore alternative optimization strategies that may allow us to leverage existing theoretical support and attempt to derive convergence results. While the current algorithm represents an initial practical approach grounded in our theoretical framework, we believe that investigating other techniques with more established guarantees represents an important research direction.

---

> ### Author Response · Authors · 2023-11-17
> **Response to Reviewer yNQQ (part 2)**
>
> **Related work on opponent modelling.** Your suggestion to include related works on opponent modeling is well-received. Indeed, our approach shares some similarities with opponent modeling in the context of solving other agents' policies. However, the anticipations are fundamentally different from predictions or estimations of other agents’ decisions used in opponent modelling. Specifically, the anticipation of agent $i$ to agent $j$, i.e., $\pi^{ij}$, is solved to optimize agent $i$’s objective (Eq.11). The constrains in Eq.11 include the discrepancy between the anticipations $\pi^{ij}$ and the true action distribution of other agents, $\pi^{jj}$, which causes a similarity to opponent modelling. However, the discrepancy is not the primary optimization objective, but instead a constraint when maximizing the agent’s own objective. The anticipation $\pi^{ij}$ is then sent to agent j, so that when agent j maximizes its objective, it can balance agent i’s objective.
>
> For opponent modelling, agent i estimates agent j’s action or other information in order to take  them into account when solving its (agent i’s) policy. However, in our method, the anticipations is solved together with agent i’s own policy to maximize agent i’s objective. It is agent j who takes into the anticipations from agent i to solve its (agent j’s) policy.
>
> It should also be noted that agent $i$’s own policy $\pi^{ii}$ is adjusted in accordance with the discrepancy between it and $\pi^{ji}$, the anticipation of the other agent $j$ about agent $i$ policy. This constraint is taken into account for the purpose of further enforcing the agreement between what is locally learned and what other anticipates, which is not the objective in opponent modeling.
>
> In the revised manuscript, we have expanded the Related work in Section 2, and included a relevant paragraph (the last paragraph in Section 4.2) to explain the above points.
>
> **Real social dilemmas and partial observability.** We believe that, in more challenging scenarios, particularly those involving dilemmas, our method's capacity to facilitate agent coordination through shared anticipations is critical. The anticipation sharing mechanism equips agents with insights into other agents’ objectives, aiding in navigating complex dilemma situations. However, we acknowledge the need for further empirical studies in these environments to validate the efficacy of our approach under such conditions.
>
> Regarding partially observable settings, our current theoretical framework is primarily oriented towards fully observable environments. Extending our approach to partially observable settings would require adaptations to account for the uncertainties and incomplete information inherent in such environments. While the core principles of anticipation sharing remain relevant, their application in partially observable scenarios would necessitate additional mechanisms to handle the lack of complete information. We recognize this as an important direction for future research to ensure the robustness of our method across a broader spectrum of multi-agent settings.
>
> **Typo in Theorem 1**.  Thanks for pointing it out. We have revised \E_s^\prime as \E_{s^\prime} in proof for Theorem 1, after Eq.20 (Eq.22 in the revised paper).

---

### Author Response · Authors · 2023-11-20
**Please provide feedback**

As the discussion period comes to a close, we kindly request your feedback regarding our rebuttal. We value your input on whether our response effectively addressed your concerns and if there are any remaining questions or points you would like us to elaborate on for further clarity.

---

### Meta-Review · Area_Chair_pm2i · 2023-12-06

**Metareview:**

This paper introduces a decentralized multi-agent reinforcement learning method that enables multi-agent coordination by the proposed anticipation sharing. Compared to sharing rewards, values, or full policies, exchanging anticipations incurs less overhead.  Experiments show anticipation sharing performs favorably over other sharing methods.

The main weaknesses are as follows:
- As it requires information exchange during training, the proposed method is semi-centralized learning. In such a case, it is desirable to show the proposed method can avoid local Nash equilibria, for example, in well-known dilemmas. However, this paper does not touch on this essential issue.
- The experimental environment is overly simplistic, which cannot provide enough evidence for the method's scalability and robustness.
- There is no convergence proof. As far as I know, the referenced fully decentralized methods (no communication), such as DPO, indeed have convergence proof. Thus, the convergence proof is needed.

**Justification For Why Not Higher Score:**

The paper has major weaknesses unaddressed.

**Justification For Why Not Lower Score:**

N/A

---

### Decision · Program_Chairs · 2024-01-16

Reject